# Explore More, Learn Better: Parallel MLLM Embeddings under Mutual Information Minimization

## Abstract

Embedding models are a cornerstone of modern AI. Driven by Multimodal Large Language Models (MLLMs), they have made great progress in architecture and data curation, while the holistic paradigm is still limited to SSC, *i.e.*, single input, singular embedding, contrastive supervision, which collapses rich, multifaceted inputs into monolithic embeddings and fails to fully exploit MLLM capabilities. In this paper, we tailor one **P**arallel **D**ecoupling **F**ramework (PDF) for multimodal embedding learning, by utilizing the proprietary steerability of MLLMs, *i.e.*, their ability to flexibly generate quite differentiated response under explicit instructions. Concretely, PDF conditions a shared MLLM backbone on distinct, learnable prefixes to roll out multiple parallel paths for one input, then relies on these paths to obtain parallel embeddings. To promote full parallel diversity, we employ Mutual Information Minimization (MIM) as an explicit constraint, coupled with per-path contrastive supervision to maintain semantic alignment. Such dual-objectives force PDF to yield robust semantic coverage and a generalizable embedding space. Ultimately, the remarkable embedding space are accessible at inference via one single forward pass, incurring negligible computational overhead. We instantiate PDF on multiple MLLM backbones and prove its effectiveness on MMEB benchmark. Significant gains are consistently achieved across various resolutions and model sizes, *e.g.*, boosting the VLM2Vec-LLaVA-1.6-LR model by a remarkable +8.9% (7B), and the VLM2Vec-Qwen2VL models by +4.2% (2B) and +3.1% (7B). In terms of efficiency, our 2B model surpasses its baseline by +2.6% using only half the computational budget. Code will be available.

## 1 Introduction

Embedding models, which encode complex inputs like text and images into dense vectors, are a cornerstone of modern AI, powering applications like semantic similarity Chechik et al. (2010); Agirre et al. (2012); Marelli et al. (2014); Chen et al. (2025c), information retrieval Lin et al. (2014); Mitra et al. (2017); Karpukhin et al. (2020); Zheng et al. (2021) and Retrieval-Augmented Generation Lewis et al. (2020); Izacard & Grave (2020); Guu et al. (2020); Jin et al. (2025). To advance embedding models, previous efforts largely follow two main paths. On the data side, many studies Zhou et al. (2024); Chen et al. (2025a); Gu et al. (2025); Lan et al. (2025) explored labor-intensive hard-sample mining. On the architectural side, early CLIP-like models, *e.g.*, UniIR on the M-BEIR benchmark Lan et al. (2025) have evolved into recent MLLM-based models like VLM2Vec on the MMEB benchmark Jiang et al. (2024b), as Multimodal Large Language Models show impressive gains over small-scale counterparts. However, from the holistic perspective, most methods still converge on one ubiquitous **SSC** paradigm shown in Fig. 1 (a): mapping a *Single input to a Singular embedding, and learning via a Contrastive supervision*. Such an identical paradigm appears increasingly outdated, mainly for two reasons. *Architectural Mismatch*: originating from simpler CLIP-like dual-encoders, SSC fails to leverage the advanced capabilities of MLLMs. *Information Bottleneck*: by collapsing a multifaceted input into a singular point in the embedding space, SSC incurs severe information loss, resulting in limited semantic coverage and reduced robustness.

This status quo drives us to tailor a proprietary embedding pipeline for MLLMs, enabling the learned embeddings to fully encompass semantic richness of input. Here, we are motivated by ***the distinctive***

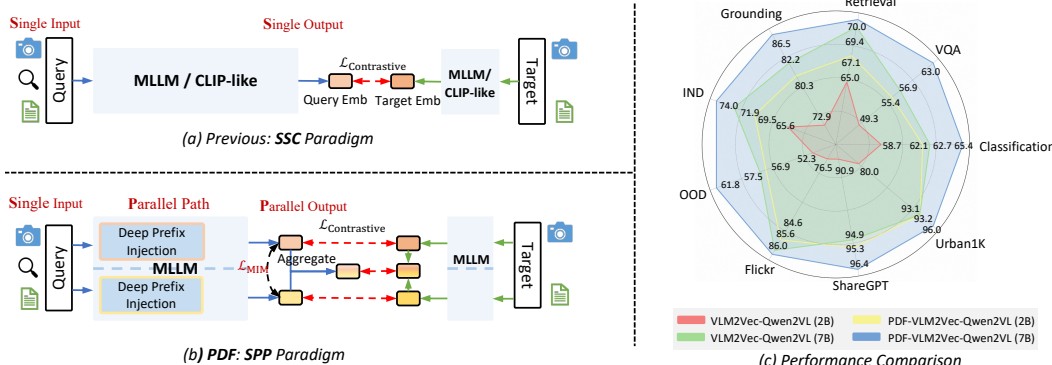

Figure 1: **Comparisons of Framework & Performance.** (a) Previous: the ubiquitous SSC frame-work, *i.e.*, single input, singular embedding, contrastive supervision. (b) Parallel Decoupling Frame-work (PDF): single input rolls out as parallel inputs, then generates multiple embeddings explicitly de-correlated by Mutual Information Minimization (MIM). (c) On the VLM2Vec-Qwen2VL back-bone, our PDF consistently delivers significant gains across diverse tasks and model scales.

*steerability* of MLLMs Liu et al. (2023; 2024a); Wang et al. (2024); Qwen et al. (2025) compared to CLIP-like dual-encoders, *i.e.*, MLLMs are highly flexible to prompts, tend to generate sufficiently differentiated response under explicit instructions. Prior work Jiang et al. (2024b) mainly leverages such steerability for cross-task adaptation with the notable success. In contrast, we creatively utilize it for fundamental embedding learning. Concretely, the model is conditioned on distinct, learnable prefixes to roll out a single input into multiple parallel prompts, which are subsequently processed by MLLMs to yield multiple parallel embeddings. This constitutes our novel **SPP** paradigm: *Single input, Parallel paths, Parallel outputs*. For the same input, prefixes are initialized differently, leading parallel rollouts to follow divergent convergences, thus achieving robust semantic coverage.

To better realize our **SPP**, naively relying on randomly initialized prefixes to *implicitly* guide parallel paths is insufficient and suboptimal. Although prefixes differ at initialization, the shared MLLM backbone and mere one strong objective (contrastive loss) create a clear tendency for the model to gradually disregard differentiation during convergence, causing all parallel paths to collapse into somehow similar, *i.e.*, redundant embeddings. To counteract this, we argue that an *explicit* constraint on parallel differentiation is necessary, *i.e.*, actively measure then penalize statistical dependencies between parallel paths. This force embeddings to become diverse, maximizing semantic coverage to minimize information loss, moving beyond the singular-embedding limitation of **SSC**.

As illustrated in Fig. 1 (b), we hence propose one novel **Parallel Decoupling Framework (PDF)**, with threefold designs. First, to instantiate the parallel paths, we employ one ***deep prefix injection*** mechanism: for each path, one unique set of learnable parameters is injected into every transformer layer, directly modulating the self-attention computation. Second, to fulfill the need for an explicit diversity constraint, we employ ***Mutual Information Minimization*** (**MIM**) Kinney & Atwal (2014). As the true data distribution is unknown, direct MI computation is intractable. We therefore mini-mize one tractable variational upper bound, implemented using the vCLUB estimator Cheng et al. (2020). This establishes a two-stage optimization game within each training step: a parametric MI estimator first learns to detect the dependency between parallel embeddings, after which the main MLLM is trained to produce more independent embeddings that "fool" this fixed estimator. Third, to ensure this diversity does not degrade the embedding quality, a standard contrastive loss is applied to each path to anchor it to semantics of the input. This dual-objective training acts as a powerful regularizer on the shared model backbone Zhang et al. (2018). Empirically, we find that a single for-ward pass at inference is sufficient to unlock the model's enhanced embedding space, *i.e.*, remaining efficient with negligible additional computational overhead.

To evaluate the effectiveness and generality, we instantiate PDF on top of the VLM2Vec paradigm, using both LLaVA-1.6 Liu et al. (2024a) and Qwen2VL Wang et al. (2024) as backbones. Exten-sive experiments on the MMEB benchmark consistently demonstrate significant performance gains across different base models, parameter scales, and learning resolutions. Notably, PDF boosts the VLM2Vec-LLaVA-1.6 (7B) model by a remarkable **+8.9** points on the low-resolution setting. While on the Qwen2VL backbone, it improves the 2B and 7B models by **+4.2** and **+3.1** points, respectively. Beyond sheer performance, PDF also demonstrates dramatic training efficiency. With only half the total computational budget, our 2B model surpasses the fully-computational baseline by **+2.6** points.

## 2 RELATED WORK

**Multimodal Large Language Models** (MLLMs) have empowered LLMs with sophisticated visual understanding. A pioneering work, LLaVA Liu et al. (2023), established a paradigm by projecting features from a pre-trained vision encoder (e.g., CLIP Radford et al. (2021)) into the LLM's word embedding space. Following this, the field has rapidly evolved, with research focusing on enhancing visual capabilities through dynamic high-resolution processing Liu et al. (2024b) and scaling up vision encoders Chen et al. (2024b), as seen in models like the Qwen-VL series Bai et al. (2023); Wang et al. (2024). This progress has opened up new avenues for applying these powerful generalist models to specialized domains, including universal representation learning.

**Multimodal Embeddings without Large-Scale Models.** Before MLLMs, multimodal embeddings are dominated by dual-encoder architectures. Models such as CLIP Radford et al. (2021), ALIGN Jia et al. (2021), and BLIP Li et al. (2022) trained separate image and text encoders via contrastive learning on large-scale image-text pairs, excelling at tasks like zero-shot classification and text-image retrieval. Subsequent work, such as UniIR Wei et al. (2024), sought to endow these dual-encoder models with more universal retrieval capabilities by introducing a comprehensive training dataset and the MBEIR benchmark. Despite the advances, their separated-encoder design inherently struggles with tasks requiring a deeper, holistic understanding or nuanced instruction-following.

**Multimodal Embeddings with Large-Scale Models.** MLLMs' unified architecture naturally overcomes the limitations of dual-encoder models. Recent work explores to adapt MLLMs for embedding tasks, such as E5-V Jiang et al. (2024a) and VLM2Vec Jiang et al. (2024b) demonstrating state-of-the-art performance by fine-tuning MLLMs with contrastive losses on benchmarks like MMEB. LamRA Jiang et al. (2024b) employs a retrieve-then-rerank pipeline, where an MLLM-based model reranks candidates from an initial retrieval stage. MMRet Zhou et al. (2024) introduced MegaPairs, a large-scale, instruction-tuning dataset for retrieval, and showed that pre-training on this data significantly gains downstream performance. Despite the success, these methods still adhere to a paradigm of *single input, singular embedding, contrastive loss*, treating MLLM as a generator of a singular embedding for any given input. Our work challenges this ubiquitous paradigm through introducing a novel PDF framework, to explicitly explore the intrinsic semantic diversity within a single input by generating multiple, decoupled representations, one direction largely unexplored.

## 3 PDF: PARALLEL DECOUPLING FRAMEWORK

Our **P**arallel **D**ecoupling **F**ramework **(PDF)** learns multiple parallel embeddings under the guidance of Mutual Information Minimization (MIM). Fig. 2 illustrates the framework overview. Our method operates on query-target pairs, denoted as $(q, t^+)$, where each element can be a single image, a single text, or one interleaved combination of both. We formally define them as $q = (q_t, q_i)$ and $t^+ = (t_t^+, t_i^+)$, where the subscripts $t$ and $i$ denote the text and image components, respectively. A component can be absent if the input is unimodal. Following prior work Jiang et al. (2024b), we augment the query $q$ with a task-specific instruction template to form the final input $q_{inst}$:

$$q_{inst} = [\text{IMAGE\_TOKEN}] \text{ Instruct: } \{\text{task\_definition}\} \text{ Query: } \{q\}, \tag{1}$$

where {task_definition} is a placeholder for a concise description of the embedding task. Note that we do not augment the target input with task instruction, following consensus.

During training, our PDF framework creates $N$ parallel computational paths within the LLM backbone, generating distinct embeddings for each query input $q_{inst}$ and the target input $t^+$, namely $\{h_q^{(i)}\}_{i=1}^N$ and $\{h_{t+}^{(i)}\}_{i=1}^N$. This is achieved by conditioning each path on a set of learnable "prefix" parameters. These parallel embeddings, along with an aggregation module computed via lightweight MLP-softmax to generate the aggregated query embedding $h_q$ and aggregated target embedding $h_{t+}$. These embeddings are supervised by our comprehensive losses (Sec. 3.3), which integrates a contrastive objective for alignment with a MIM objective for diversity.

At inference time, the parallel mechanism and aggregation module are bypassed for efficiency, as shown by the red line in Fig. 2. We adopt only a single pre-determined path (*e.g.*, the one corresponding to the first prefix) to get final outputs, making the inference procedure identical to that of the baseline and introducing negligible additional computational overhead or latency. As a result, our method retains the benefits of enriched training without incurring any deployment cost.

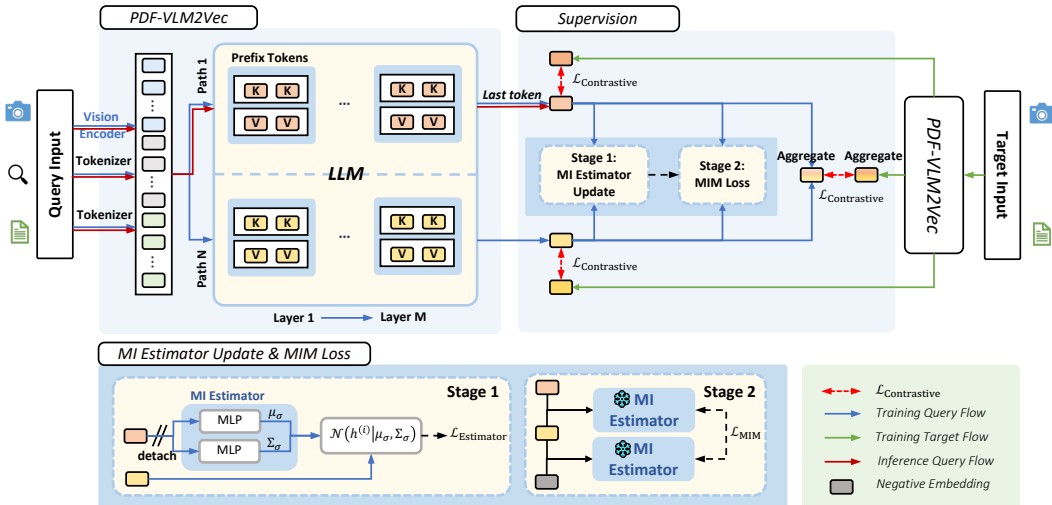

Figure 2: **Overview of the PDF-VLM2Vec Pipeline.** The training process (blue and green lines) is guided by a dual-objective system. For each input, parallel embeddings are generated via learnable prefixes. These are then supervised by: (1) Contrastive Loss to maintain representation quality, and (2) MIM Loss to enforce diversity. The MIM loss is calculated in a two-stage process: first updating the MI estimator with detached embeddings (Stage 1), and then using the frozen estimator to compute the loss (Stage 2). During inference (red line), a single forward pass inherits these benefits, yielding a robust embedding with additional negligible computational overhead.

### 3.1 GENERATING DIVERSE EMBEDDINGS VIA DEEP PREFIX INJECTION

To foster the exploration of a diverse embedding space, we introduce a deep prefix injection mechanism. Drawing inspiration from prefix-tuning Li & Liang (2021), *i.e.*, models can be steered by conditioning on different prefixes, we apply the idea at a deeper level to create our $N$ parallel computational paths. Instead of merely prepending tokens to the input sequence, we inject path-specific, learnable prefix parameters directly into each transformer layer of the LLM backbone.

We define a unique prefix that modulates the self-attention mechanism within each of the $M$ transformer layers for each of the $N$ parallel paths. For path $i$ and layer $l$, a specific prefix is denoted as $(\mathbf{p}_\mathbf{K}^{(i,l)}, \mathbf{p}_\mathbf{V}^{(i,l)}) \in (\mathbb{R}^{K \times d}, \mathbb{R}^{K \times d})$, where $K$ is the length of prefix tokens and $d$ is the dimension.

Then we use prefixes in the self-attention mechanism in LLM. Specifically, for each layer of each path, we first project the token sequence to obtain $\mathbf{Q} \in \mathbb{R}^{L \times d}$, $\mathbf{K} \in \mathbb{R}^{L \times d}$ and $\mathbf{V} \in \mathbb{R}^{L \times d}$, where the $L$ is the sequence length. Then we concatenate the $\mathbf{K}$ and $\mathbf{V}$ with their corresponding prefix:

$$\mathbf{K}' = \text{Concat}(\mathbf{p}_K^{(i,l)}, \mathbf{K}) \in \mathbb{R}^{(L+K) \times d}, \quad \mathbf{V}' = \text{Concat}(\mathbf{p}_V^{(i,l)}, \mathbf{V}) \in \mathbb{R}^{(L+K) \times d}. \quad (2)$$

Then the $\mathbf{Q}$ performs attention over augmented key-value pairs. This layer-wise modulation, repeated across the model's depth with path-specific parameters, guides each path to produce a distinct output sequence. Finally, we select the last token as the output embedding $h^{(i)}$ for path $i$.

Having obtained the parallel embeddings, one critical question then arises: ***how can we explicitly promote diversity?*** While one might hope that the distinct prefixes would naturally lead to diverse outputs Chen et al. (2025b), we contend that an explicit supervisory signal is far more effective. Therefore, we employ a Mutual Information Minimization (MIM) objective to actively discourage statistical dependence among the parallel embeddings, thereby compelling the model to discover a multifaceted embedding space that covers comprehensive semantics.

### 3.2 PROMOTING DIVERSE EMBEDDINGS VIA MUTUAL INFORMATION MINIMIZATION

To explicitly enforce diversity among the $N$ parallel embeddings $\{h^{(i)}\}_{i=1}^N$, our goal is to minimize the Mutual Information (MI) between any pair of them, $I(h^{(i)}; h^{(j)})$. However, directly computing MI is intractable as it requires access to the true data distributions. We hence minimize a tractable variational upper bound, formulated as the two-stage optimization.

**Variational Upper Bound.** To minimize a tractable upper bound of MI, we adopt the Contrastive Log-ratio Upper-Bound (CLUB) Cheng et al. (2020) estimator. CLUB introduces a variational distribution $q_\sigma(h^{(i)}|h^{(j)})$, parameterized by a neural network (***MI Estimator***) with parameters $\sigma$, to approximate the true conditional probability $p(h^{(i)}|h^{(j)})$. MI is then upper-bounded by:

$$I(h^{(i)}; h^{(j)}) \leq \mathbb{E}_{p(h^{(i)}, h^{(j)})}[\log q_\sigma(h^{(i)}|h^{(j)})] - \mathbb{E}_{p(h^{(i)})p(h^{(j)})}[\log q_\sigma(h^{(i)}|h^{(j)})]. \tag{3}$$

Minimizing the upper bound encourages embeddings $h^{(i)}$ and $h^{(j)}$ to be statistically independent.

**Two-Stage Adversarial-like Optimization.** Within each training iteration, the process unfolds as a two-stage game, where the MI Estimator and the main MLLM play opposing roles.

• *Training the MI Estimator.* First, we train the MI Estimator to detect statistical dependencies between parallel embeddings. Given a pair $(h^{(i)}, h^{(j)})$ generated from the *same* input sample (a "positive" pair), we train the estimator to predict $h^{(i)}$ from $h^{(j)}$ by maximizing the conditional log-likelihood. We parameterize $q_\sigma$ as a Gaussian, $q_\sigma(h^{(i)}|h^{(j)}) = \mathcal{N}(h^{(i)}|\mu_\sigma(h^{(j)}), \Sigma_\sigma(h^{(j)}))$, where an MLP predicts the mean $\mu$ and covariance $\Sigma$. The objective for the estimator's parameters $\sigma$ is:

$$\mathcal{L}_{\text{Estimator}} = -\mathbb{E}_{p(h^{(i)}, h^{(j)})}[\log q_\sigma(h^{(i)}|h^{(j)})]. \tag{4}$$

Crucially, during this stage, the gradients from this loss only update $\sigma$; the embeddings $h^{(i)}$ and $h^{(j)}$ are *detached* from the computation graph of MLLMs.

• *Training the MLLM backbone.* Second, with the MI Estimator's parameters $\sigma$ *frozen*, we update the MLLM parameters $\theta$. The goal now is to make the estimator's job harder by generating embeddings that are less predictable. We do this by minimizing the MI upper bound from Eq. (3), which serves as our MIM loss. For a mini-batch of size $B$, $\mathcal{L}_{\text{MIM}}$ is approximated as:

$$\mathcal{L}_{\text{MIM}} = \frac{1}{B} \sum_{k=1}^{B} \frac{1}{N(N-1)} \sum_{i \neq j} \left( \log q_\sigma(h_k^{(i)}|h_k^{(j)}) - \mathbb{E}_{m \neq k}[\log q_\sigma(h_k^{(i)}|h_m^{(j)})] \right). \tag{5}$$

Here, the first term is the log-likelihood for a "positive" pair from the same sample $k$. The second term is the expectation over "negative" pairs, approximated by pairing $h_k^{(i)}$ with an embedding $h_m^{(j)}$ from another sample $m$ within the same batch. The total gradient from $\mathcal{L}_{\text{MIM}}$ (and the contrastive loss) then updates $\theta$, pushing the parallel embeddings towards greater differentiation.

## 3.3 OVERALL TRAINING OBJECTIVE

Our training objective comprises two crucial components: one contrastive loss $\mathcal{L}_{\text{contrastive}}$ to enforce representation quality, and one Mutual Information Minimization loss $\mathcal{L}_{\text{MIM}}$ to promote diversity.

**Contrastive Loss.** Our primary supervision signal is the InfoNCE loss Oord et al. (2018), which enforces query-target semantic alignment. We first apply it to the aggregated embeddings: for a given query's aggregated embedding $h_q$, its corresponding aggregated target embedding $h_{t^+}$ serves as the positive sample, while all other target embeddings in the mini-batch $\mathcal{B}^-$ act as negatives:

$$\mathcal{L}_{\text{InfoNCE}}(h_q, h_{t^+}) = -\log \frac{\exp(\text{sim}(h_q, h_{t^+})/\tau)}{\exp(\text{sim}(h_q, h_{t^+})/\tau) + \sum_{h_{t^-} \in \mathcal{B}^-} \exp(\text{sim}(h_q, h_{t^-})/\tau)}, \tag{6}$$

where $\text{sim}(\cdot, \cdot)$ is cosine similarity and $\tau$ is a temperature hyperparameter.

To prevent the diversity encouraged by $\mathcal{L}_{\text{MIM}}$ from degrading representation quality, we also apply the InfoNCE loss to each of the $N$ parallel paths. This serves as a crucial representation constraint, compelling each path to learn semantically valid representations rather than collapsing into several trivial solutions (*e.g.*, random noise) to minimize mutual information. Therefore, the contrastive loss supervises both aggregated and parallel embeddings, by a weighting hyperparameter $\lambda_{\text{CON}}$.

$$\mathcal{L}_{\text{contrastive}} = \mathcal{L}_{\text{InfoNCE}}(h_{q,\text{agg}}, h_{t^+}) + \lambda_{\text{CON}} \frac{1}{N} \sum_{i=1}^{N} \mathcal{L}_{\text{InfoNCE}}(h_q^{(i)}, h_{t^+}^{(i)}), \tag{7}$$

where $h_{t^+}^{(i)}$ is the $i$-th parallel embedding of the positive target.

**Total Training Objective.** Finally, our PDF framework is optimized through a linear combination of the contrastive loss for quality and the MIM loss (from Sec. 3.2) for diversity:

$$\mathcal{L}_{\text{total}} = \mathcal{L}_{\text{contrastive}} + \lambda_{\text{MIM}} \mathcal{L}_{\text{MIM}}. \tag{8}$$

# 4 EXPERIMENT

## 4.1 DATASETS & METRICS & IMPLEMENTATIONS

**Datasets & Metrics.** Following VLM2Vec Jiang et al. (2024b), we train on the 20 in-distribution MMEB datasets covering 662K pairs across four meta-tasks: classification, VQA, multimodal retrieval, and visual grounding. The model is then evaluated on both 20 in-distribution and 16 out-of-distribution MMEB test sets. We report Precision@1 as metrics on each dataset, *i.e.*, the proportion of top-ranked candidates that are positive samples.

**Implementation Details.** We instantiate PDF upon the VLM2Vec Jiang et al. (2024b), with multiple backbones, model scales and resolutions. For LLaVA-1.6 (7B), we train both a low-resolution (LR, 334x334) and a high-resolution (HR, 1344x1344) variant, referred to PDF-VLM2Vec-LLaVA1.6-LR and PDF-VLM2Vec-LLaVA1.6. For Qwen2VL, we train HR versions for both 2B and 7B scales, as well as an LR version (LR, 128x128) for the 2B model. These are denoted as PDF-VLM2Vec-Qwen2VL (for HR) and PDF-VLM2Vec-Qwen2VL-LR (for the 2B LR). For efficient fine-tuning, we apply LoRA Hu et al. (2022) to the LLM backbone with a rank of $r = 8$ in all experiments.

We set the number of parallel paths to $N = 2$. Each path is conditioned by a deep prefix of length $K = 20$ injected into each transformer layer. The aggregated representation is computed through one lightweight MLP, which consists of two linear layers with a SiLU activation function. Besides, for the loss weight hyperparameters, we set $\lambda_{\text{MIM}} = 1 \times 10^{-4}$ and $\lambda_{\text{CON}} = 1.0$. These hyperparameters are determined based on ablation studies detailed in Appendix A.5.

All models are trained with a global batch size of 1024 and an InfoNCE temperature of $\tau = 0.02$. To support this large batch size, we leverage GradCache Jiang et al. (2024b), following the original VLM2Vec implementation. We use the AdamW optimizer Loshchilov & Hutter (2017) with a peak learning rate of $2 \times 10^{-5}$ for the 2B models and $5 \times 10^{-6}$ for the 7B model. Besides, the training schedule consists of 2000 steps, including a 100-step linear warm-up followed by a linear learning rate decay. All experiments were conducted on NVIDIA H100 GPUs.

**Baselines.** We evaluate our method against two main categories of baselines. Our primary and most direct competitors are the **VLM2Vec** Jiang et al. (2024b) models, the state-of-the-art framework upon which our work is built. To ensure a fair and direct comparison, for each of our PDF-enhanced models, we train its corresponding VLM2Vec counterpart using the identical dataset, hyperparameters, and overall training procedure. This strictly controlled setup allows us to isolate the performance gains attributable specifically to our PDF framework. Secondly, to contextualize our results within the broader landscape of multimodal representation learning, we also compare against a wide range of established models. Following the evaluation protocol from VLM2Vec, this group includes both prominent LLM-based and non-LLM-based methods. Unless otherwise specified, all baseline results are taken directly from original papers. The only exception is the VLM2Vec-Qwen2VL-LR baseline, which we reproduced under the same controlled settings for a direct comparison.

## 4.2 MAIN RESULTS

● *Comparisons with SOTA on the MMEB benchmark.* Table 1 summarizes the performance of our PDF-VLM2Vec against various baselines. The direct baseline, VLM2Vec-Qwen2VL, is shaded in gray, while the absolute gains ($\Delta$) of our method are highlighted in blue.

The results unequivocally demonstrate the broad effectiveness and generalizability of our PDF training strategy. Our method consistently outperforms the VLM2Vec baselines, across different model scales, data resolutions and foundational MLLM. For the primary Qwen2VL baseline, PDF-VLM2Vec achieves substantial overall improvements of **+12.1**, **+4.2**, and **+3.1** points for the low-res 2B, high-res 2B, and high-res 7B models, respectively. The versatility of our strategy is further evidenced by its application to VLM2Vec-LLaVA-1.6 (7B), where it delivers impressive gains of **+8.9** points in the low-resolution setting and **+1.8** points in the high-resolution setting. This consistent performance enhancement across two distinct VLM architectures strongly validates the scalability and universal applicability of our proposed method.

● *Zero-Shot Image/Text Retrieval.* To further validate generalization, we conduct zero-shot retrieval experiments on three unseen datasets: Flickr30K Plummer et al. (2015), ShareGPT4V Chen et al.

Table 1: **Comparisons with SOTA on the MMEB benchmark.** Our PDF-VLM2Vec is evaluated against both non-LLM and LLM-based baselines. Scores are averaged per meta-task and reported for In-Distribution (IND), Out-of-Distribution (OOD), and Overall performance. The direct baselines, VLM2Vec, are shaded in gray; $\Delta$ denotes the absolute improvement over the direct baseline.

| Model | Per Meta-Task Score | | | | Average Score | | |
|---|---|---|---|---|---|---|---|
| | Class. | VQA | Retrieval | Grounding | IND | OOD | Overall |
| # of datasets → | 10 | 10 | 12 | 4 | 20 | 16 | 36 |
| *No LLM based Method* | | | | | | | |
| CLIP Radford et al. (2021) | 42.8 | 9.1 | 53.0 | 51.8 | 37.1 | 38.7 | 37.8 |
| BLIP2 Li et al. (2023) | 27.0 | 4.2 | 33.9 | 47.0 | 25.3 | 25.1 | 25.2 |
| SigLIP Zhai et al. (2023) | 40.3 | 8.4 | 31.6 | 59.5 | 32.3 | 38.0 | 34.8 |
| OpenCLIP Cherti et al. (2023) | 47.8 | 10.9 | 52.3 | 53.3 | 39.3 | 40.2 | 39.7 |
| UniIR(CLIP_SF) Wei et al. (2024) | 44.3 | 16.2 | 61.8 | 65.3 | 47.1 | 41.7 | 44.7 |
| CLIP-FFT Jiang et al. (2024b) | 55.2 | 19.7 | 53.2 | 62.2 | 47.6 | 42.8 | 45.4 |
| OpenCLIP-FFT Jiang et al. (2024b) | **56.0** | **21.9** | **65.4** | **64.1** | **50.5** | **43.1** | **47.2** |
| *LLM-based model (2B model)* | | | | | | | |
| ColPali v1.3 Faysse et al. (2024) | 40.3 | 11.5 | 48.1 | 40.3 | - | - | 34.9 |
| GME Zhang et al. (2024b) | 54.4 | 29.9 | 66.9 | 55.5 | - | - | 51.9 |
| VLM2Vec-Qwen2VL-LR | 51.9 | 29.6 | 54.9 | 50.6 | 50.1 | 41.5 | 46.5 |
| PDF-VLM2Vec-Qwen2VL-LR | 59.3 | 47.5 | 63.3 | 70.0 | 62.8 | 53.3 | 58.6 |
| $\Delta$ - baseline | +7.4 | +17.9 | +8.4 | +19.4 | +12.7 | +11.8 | +12.1 |
| VLM2Vec-Qwen2VL | 58.7 | 49.3 | 65.0 | 72.9 | 65.6 | 52.3 | 59.7 |
| PDF-VLM2Vec-Qwen2VL | **62.1** | **55.4** | **67.1** | **80.3** | **69.5** | **56.9** | **63.9** |
| $\Delta$ - baseline | +3.4 | +6.1 | +2.1 | +7.4 | +3.9 | +4.6 | +4.2 |
| *LLM-based model (7B model)* | | | | | | | |
| GME Zhang et al. (2024b) | 57.7 | 34.7 | **71.2** | 59.3 | - | - | 56.0 |
| LamRA-Qwen2 Liu et al. (2025) | 59.2 | 26.5 | 70.0 | 62.7 | - | - | 54.1 |
| LamRA-Qwen2.5 Liu et al. (2025) | 51.7 | 34.1 | 66.9 | 56.7 | - | - | 52.4 |
| VLM2Vec-LLAVA-1.6-LR | 54.7 | 50.3 | 56.2 | 64.0 | 61.0 | 45.7 | 55.0 |
| PDF-VLM2Vec-LLAVA-1.6-LR | 58.7 | 55.2 | 67.5 | 88.1 | 69.8 | 56.7 | 63.9 |
| $\Delta$ - baseline | +4.0 | +4.9 | +11.3 | +24.1 | +8.8 | +11.0 | +8.9 |
| VLM2Vec-LLAVA-1.6 | 61.2 | 49.9 | 67.4 | 86.1 | 67.5 | 57.1 | 62.9 |
| PDF-VLM2Vec-LLAVA-1.6 | 59.7 | 56.1 | 67.8 | **89.2** | 70.4 | 57.5 | 64.7 |
| $\Delta$ - baseline | -1.5 | +6.2 | +0.4 | +3.1 | +2.9 | +0.4 | +1.8 |
| VLM2Vec-Qwen2VL | 62.7 | 56.9 | 69.4 | 82.2 | 71.9 | 57.5 | 65.5 |
| PDF-VLM2Vec-Qwen2VL | **65.4** | **63.0** | 70.0 | 86.5 | **74.0** | **61.8** | **68.6** |
| $\Delta$ - baseline | +2.7 | +6.1 | +0.6 | +4.3 | +2.1 | +4.3 | +3.1 |

(2024a), and Urban1K Zhang et al. (2024a). As shown in Table 4.2, our PDF framework delivers compelling performance gains. The improvements are particularly dramatic for the 2B model, which sees performance boosts of up to **+17.2** points on Urban1K. Moreover, our method consistently improves upon the already strong 7B baseline across all benchmarks. These results demonstrate that our approach not only substantially enhances the generalization of smaller models but also robustly scales to larger, more capable ones, confirming its broad effectiveness.

## 4.3 ABLATION STUDY

Table 3 conducts comprehensive ablation studies, structured in two parts: validating the effectiveness of components and evaluating different inference strategies. All experiments are performed on the high-resolution MMEB dataset using VLM2Vec-Qwen2VL (2B).

● *Effectiveness of Training Components.* The top section of the table illustrates the step-by-step construction of our model. We begin with the VLM2Vec baseline (R1, 59.7 Overall). First, to verify that our gains do not simply stem from increased parameterization, we add prefix parameters to the baseline in a single-path setting (R2). This yields no performance improvement, confirming that the parallel architecture is the true source of gains. Indeed, introducing **Parallel Paths** (R3) provides a significant +2.4 point uplift. Subsequently, applying the **MIM Loss** (R4) to encourage diversity brings a further +1.2 point improvement. Finally, adding the **Sub Loss** to enforce quality on each

Table 2: **Zero-shot image-text retrieval on the unseen datasets: Flickr30K, ShareGPT4V and Urban1K.** Recall@1 (R@1) scores are reported. For both 2B and 7B model scales, $\Delta$ denotes the absolute point improvement over the direct baseline.

| Model | Text - Image (R@1) | | | Image - Text (R@1) | | |
|---|---|---|---|---|---|---|
| | Flickr30K | ShareGPT4V | Urban1K | Flickr30K | ShareGPT4V | Urban1K |
| CLIP | 79.5 | 90.1 | 77.8 | 92.9 | 93.6 | 80.7 |
| EVA-CLIP-8B | 80.3 | 93.1 | 80.4 | 94.5 | 91.2 | 77.8 |
| E5-V (7B) | 77.3 | 85.1 | 88.9 | 85.7 | 82.1 | 83.2 |
| LAMRA-RET (7B) | 82.8 | 93.3 | 95.1 | 92.7 | 88.1 | 94.3 |
| VLM2Vec-LLAVA1.6 (7B) | 76.0 | 85.8 | 84.7 | 90.6 | 90.7 | 90.8 |
| PDF-VLM2Vec-LLAVA1.6 (7B) | 80.0 | 87.8 | 90.5 | 93.9 | 93.5 | 91.7 |
| $\Delta$ - **baseline** | +4.0 | +2.0 | +5.8 | +3.3 | +2.8 | +0.9 |
| VLM2Vec-Qwen2VL (2B) | 68.4 | 89.4 | 75.5 | 84.5 | 92.1 | 84.4 |
| PDF-VLM2Vec-Qwen2VL (2B) | 77.1 | 94.7 | 92.7 | 92.0 | 95.8 | 93.6 |
| $\Delta$ - **baseline** | +8.7 | +5.3 | +17.2 | +7.5 | +3.7 | +9.2 |
| VLM2Vec-Qwen2VL (7B) | 79.6 | 93.0 | 92.9 | 91.6 | 96.7 | 93.2 |
| PDF-VLM2Vec-Qwen2VL (7B) | 79.7 | 95.3 | 96 | 92.3 | 97.4 | 95.9 |
| $\Delta$ - **baseline** | +0.1 | +2.3 | +3.1 | +0.7 | +0.7 | +2.7 |

Table 3: **Ablation study of all components.** We start from the baseline (R1) and incrementally add deep prefix tunning (Prefix, R2), Parallel Embeddings (Parallel, R3), MIM Loss (MIM, R4), and Subspace Contrastive Loss (Sub Loss, R5). R6-R7 compares alternative inference strategies.

| ID | Module | | | | | Average Score | | |
|---|---|---|---|---|---|---|---|---|
| | Prefix | Parallel Path | MIM | Sub Loss | Inference Strategy | IND | OOD | Overall |
| R1 | - | - | - | - | - | 65.6 | 52.3 | 59.7 |
| R2 | ✓ | - | - | - | Single Prefix | 64.9 | 53.2 | 59.7 |
| R3 | ✓ | ✓ | - | - | Single Prefix | 67.2 | 55.8 | 62.1 |
| R4 | ✓ | ✓ | ✓ | - | Single Prefix | 68.0 | **57.4** | 63.3 |
| **R5** | ✓ | ✓ | ✓ | ✓ | Single Prefix | **69.6** | 56.9 | **63.9** |
| R6 | ✓ | ✓ | ✓ | ✓ | Aggregate | 69.5 | 56.9 | **63.9** |
| R7 | ✓ | ✓ | ✓ | ✓ | No Prefix | 33.4 | 25.0 | 29.7 |

path (R5) results in our full model, achieving the best performance of 63.9 Overall. This incremental improvement at each step validates the efficacy of our core design choices.

• *Analysis of Inference Strategies.* The bottom section of the table investigates different inference strategies, all applied to the same fully-trained model (R5). Most critically, removing the prefix during inference (R7) leads to a catastrophic performance collapse (29.7 Overall), demonstrating that the learned prefixes are essential for activating the correct representational subspaces. Furthermore, we observe that using the aggregated embedding (R6) yields identical performance to our proposed strategy of using a single prefix path (R5). Given that the single-path approach incurs almost the same computation overload compared to the baseline (Table 4), which is ideal for deployment. Note that test results generated using different path prefixes are essentially identical.

## 4.4 ANALYSIS OF EFFICIENCY AND DIVERSITY

We conduct a comprehensive analysis to demonstrate that PDF's performance gains are achieved with remarkable efficiency, both in terms of computation and parameters.

• *Parameter & Inference Efficiency.* As detailed in Table 4, the performance gains are not a byproduct of increased parameters. Baseline VLM2Vec with a doubled LoRA rank (r=16) actually shows a performance drop, whereas our PDF, with a comparable number of trainable parameters (0.449% *vs.* 0.415%), delivers one substantial +7.9% improvement (63.9 *vs.* 56.0). Most critically, our standard "Single Prefix" inference incurs virtually no additional computational cost (18.937 *vs.* 18.925 TFLOPs), confirming that the benefits of our diversified training are inherited at zero overhead.

Table 4: **Inference efficiency of PDF.** We analyze the parameter and computational overhead of PDF on the VLM2Vec-Qwen2VL (2B) model. Inference TFLOPs are calculated using one 1344x1344 image. The table highlights that our standard inference strategy ("Single Prefix") achieves a significant +4.2 point performance gain over the baseline with negligible additional computational overhead (**+0.06%**) and only one minor increase in trainable parameters.

| Model | Inference | Params (B) | Trainable (%) | TFLOPs | MMEB Score |
|---|---|---|---|---|---|
| VLM2Vec (LoRA, r=8) | - | 2.214 | 0.211 | 18.925 | 59.7 |
| VLM2Vec (LoRA, r=16) | - | 2.218 | 0.415 | 18.925 | 56.0 |
| PDF-VLM2Vec | Aggregate | 2.219 | 0.449 | 24.999 | 63.9 |
|  | **Single Prefix** |  |  | **18.937** | **63.9** |

Table 5: **Training efficiency of PDF.** Our PDF-VLM2Vec demonstrates significant improvements in the training efficiency compared to the VLM2Vec baseline.

| Model | Iters | 2B Model | | | 7B Model | | |
|---|---|---|---|---|---|---|---|
|  |  | IND | OOD | Overall | IND | OOD | Overall |
| VLM2Vec-Qwen2VL | 2000 | 65.6 | 52.3 | 59.7 | 71.9 | 57.5 | 65.5 |
| PDF-VLM2Vec-Qwen2VL | 500 | 66.4 | 57.3 | 62.3 (+2.6) | 70.8 | 61.2 | 66.5 (+1.0) |
|  | 1000 | 68.3 | **57.5** | 63.5 (+3.8) | 72.8 | 60.2 | 67.1 (+1.6) |
|  | 2000 | **69.5** | 56.9 | 63.9 (+4.2) | **74.0** | **61.8** | **68.5** (+3.0) |

● *Training Efficiency & Convergence.* While parallel training of PDF increases computation per iteration (approximately 1.32 times, comparing Aggregate *vs.* baseline TFLOPs), this is overwhelmingly compensated by accelerated convergence. As shown in Fig. 3 Left, our 2B model surpasses the final performance of the fully-trained baseline (59.7) in just **500 iterations**, achieving a score of 62.3 as shown in Table 5. This indicates our model reaches one superior state using only **50% of the total computational budget** (500 iters $\times$ 2 paths *vs.* 2000 iters $\times$ 1 path), highlighting a more effective learning process.

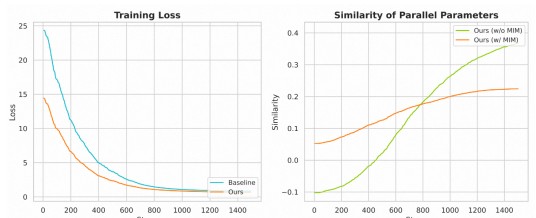

Figure 3: **Record of the training dynamics on VLM2Vec-Qwen2VL (2B). Left**: Training loss *vs.* baseline. **Right**: Cosine similarity of parallel prefixes w/ and w/o MIM loss.

● *Visualizing Diversity.* Fig. 3 Right reveals the mechanism behind this efficiency. Without MIM constraint, these parallel prefixes' embeddings quickly collapse to similar. In contrast, this collapse is effectively counteracted with the MIM loss, enforcing diversity throughout training. We hypothesize this enforced diversity compels the model to explore a wider representational space, avoiding suboptimal local minima and accelerating the discovery of a robust, generalizable solution.

## 5 CONCLUSION

This paper introduces PDF, a novel VLM2Vec training framework designed to enhance the robustness and efficiency of embedding learning. Our core idea is to guide MLLMs to encourage diverse embedding exploration through parallel paths supervised by a Mutual Information Minimization (MIM) objective. During inference, we find the embedding space can be effectively activated by a single path, resulting in significant performance gains with negligible additional computational overhead. Extensive experiments on the MMEB benchmark and zero-shot benchmark validate the effectiveness across various tasks and model setting. We demonstrated that integrating PDF into the VLM2Vec framework not only substantially improves performance across a wide range of tasks and model scales but also accelerates training convergence. We hope that this work can inspire researchers to explore more effective training frameworks for the MLLM-based embedding models.

**Ethics Statement** This work has been conducted in accordance with the ICLR Code of Ethics. We have carefully considered the ethical implications of our research and can confirm several key points. All training and evaluation datasets in this study are publicly available and have been previously peer-reviewed.

**Reproducibility Statement** To ensure the reproducibility of our work, we will ensure the following points. **Code:** Our code and model will be made publicly available, including necessary scripts. **Data:** All the training datasets and evaluation datasets are publicly available. **Experimental Setup:** We have stated all experimental configurations, including hyperparameters, hardware specifications in the Training Details of the main paper. **Model Architecture:** The architecture details and training procedures are thoroughly described in method part, with additional technical specifications provided in the appendix.

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

# A APPENDIX

## A.1 ANALYSIS OF VCLUB ESTIMATOR STABILITY AND VARIANCE

A potential concern with any method relying on variational inference is the stability and potential bias of the estimator. To address this, we empirically investigate the stability of the vCLUB Mutual Information (MI) estimator used in our PDF framework. We conducted four independent training runs of the PDF-VLM2Vec-Qwen2VL-LR model. Each run used a different random seed, which affects parameter initialization and data shuffling, to ensure a robust assessment of stability.

We analyzed the training dynamics of the MIM loss, which is directly computed using the vCLUB estimator. As illustrated in Figure 4 (a), the MIM loss curves across the four independent runs are highly consistent. The shaded area, representing the standard deviation across runs, is extremely narrow, indicating that the estimator provides a stable and reliable signal to the main model throughout the training process. This suggests that the two-stage optimization game between the MI estimator and the MLLM backbone converges without significant oscillations or instability caused by the estimator.

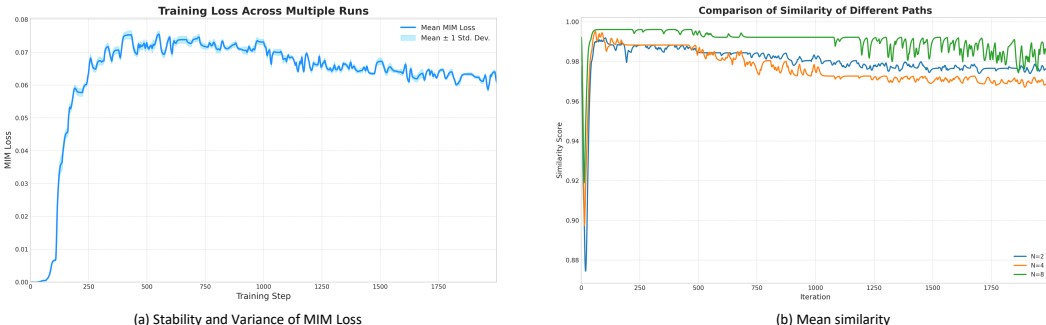

(a) Stability and Variance of MIM Loss
(b) Mean similarity

Figure 4: **Training Stability and Path Diversity.** (a) The MIM loss across multiple runs shows minimal variance (shaded area), demonstrating the stability of the vCLUB estimator. (b) The $N = 8$ model (green) maintains higher path similarity, indicating less effective decoupling than $N = 2$ (blue) and $N = 4$ (orange).

## A.2 ATTENTION ANALYSIS OF DIFFERENT PATHS

A key finding in our main paper is that while our PDF framework successfully induces diversity, aggregating the parallel paths at inference does not yield significant performance gains over a single, well-trained path. This section provides a qualitative analysis via attention visualization to explain this phenomenon. It shows that the primary role of diversity in our framework is to act as a training-time regularizer, rather than to produce complementary experts for inference-time aggregation.

To provide strong evidence for this regularization view, we visualized the last layer mean attention from the last token for two parallel paths processing the same input sample, as shown in Figure 5. This visualization provides two critical insights:

**Convergence on Core Semantics:** As shown in Figure 5 (Top, Middle), the attention distributions for Path 1 and Path 2 are macroscopically similar. Both paths allocate significant attention to the same token groups, confirming they have learned to focus on the same core semantic content as required by the primary contrastive loss. This explains why each individual path becomes a high-performing expert and why their final embeddings exhibit high cosine similarity, as they are both grounded in the same fundamental understanding of the input.

**Diversity as Subtle Shifts in Focus:** The role of the MIM objective becomes clear when examining the difference map (Figure 5, Bottom). The non-zero differences, are distributed across many heads. This demonstrates that MIM successfully prevents the paths from becoming identical. This "diversity" manifests as slight variations in how much attention each head pays to a specific token. Instead of learning orthogonal semantic concepts, our framework forces the paths to "view" the same concepts with slightly different weighting. It is training-time difference that acts as a powerful

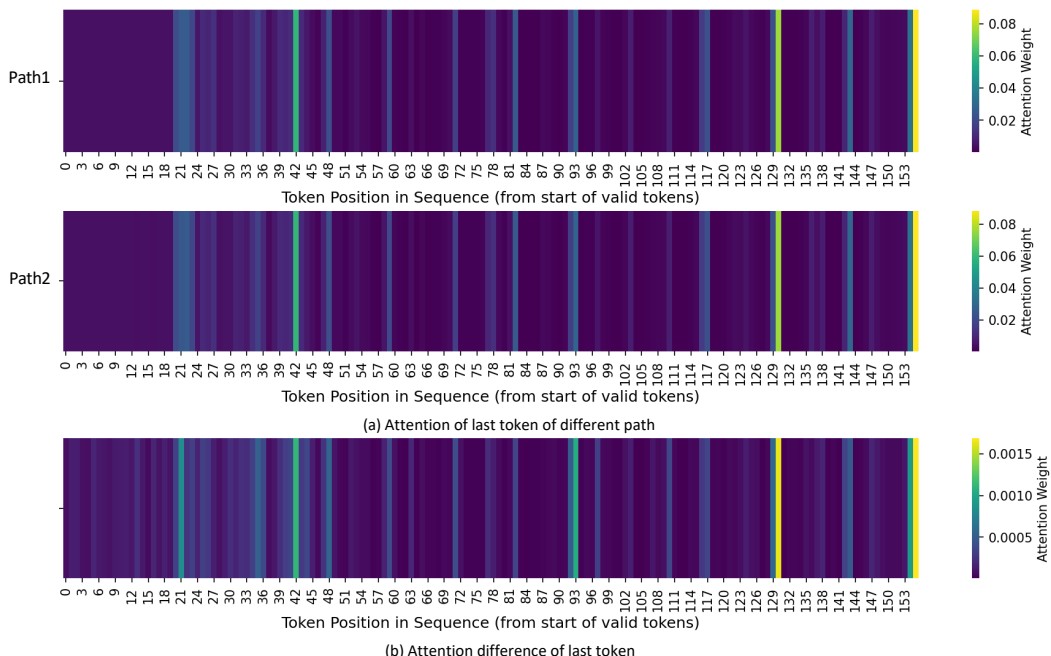

(a) Attention of last token of different path

(b) Attention difference of last token

Figure 5: **Attention map analysis of parallel paths.** We visualize the final layer attention from the last token for two parallel paths (Path 1, Path 2) processing the same input sample. **(Top, Middle)** The attention patterns for each path. The overall distributions are highly similar, indicating both paths capture the same core semantics. **(Bottom)** The absolute difference between the two attention maps. The differences across various heads illustrate the "subtle diversity" enforced by the MIM objective.

regularizer, forcing the shared backbone to develop a more robust and generalized representation space.

## A.3 ADDITIONAL ZERO-SHOT EXPERIMENTS

To further validate the generalization capabilities of our PDF framework beyond the MMEB benchmark, we conducted new zero-shot retrieval experiments. We selected two diverse, unseen datasets from distinct domains: Fashion200K Han et al. (2017) for fine-grained fashion retrieval and InfoSeek Chen et al. (2023) for document-visual question answering. The results, comparing our PDF-trained models against their direct VLM2Vec baselines, are presented in Table 6.

Table 6: **Zero-shot experiments on additional unseen datasets.** Our PDF framework demonstrates consistent and significant performance gains, highlighting its improved generalization. Note that the I and T represent Image and Text separately.

| Model | I-T | (I+T)-T | T-I | (I+T)-(I+T) |
|---|---|---|---|---|
| | Fashion200K (R@10) | InfoSeek (R@5) | Fashion200K (R@10) | InfoSeek (R@5) |
| VLM2Vec-Qwen2VL (2B) | 4.7 | 30.4 | 5.2 | 41.5 |
| PDF-VLM2Vec-Qwen2VL (2B) | 7.7 (+3.0) | 33.9 (+3.5) | 12.7 (+7.5) | 41.4 (-0.1) |
| VLM2Vec-Qwen2VL (7B) | 9.6 | 37.2 | 13.6 | 44.7 |
| PDF-VLM2Vec-Qwen2VL (7B) | 10.2 (+0.6) | 42.1 (+4.9) | 14.1 (+0.5) | 44.6 (-0.1) |

## A.4 COMPARISON WITH PROMPT-BASED METHODS

To contextualize our framework with respect to prompt-based methods for eliciting embeddings, we conduct a comparative experiment against a baseline inspired by MetaEoL Lei et al. (2024).

**Experimental Setup:** We establish a VLM2Vec-MetaEoL baseline by applying the MetaEoL inference strategy to the pre-trained VLM2Vec-Qwen2VL-2B model. For each input, we generate 8 embeddings using the 8 distinct prompts from the original MetaEoL paper and average them to produce the final embedding. This strategy is also applied to our trained PDF model for a comprehensive comparison. All experiments are conducted on the MMEB benchmark.

**Results and Discussion:** The results, as presented in Table 7, highlight a fundamental distinction between training-time frameworks like ours and inference-time prompting strategies. Key observations from this comparison are:

- **Ineffectiveness of Post-Hoc Prompting:** Applying MetaEoL prompts to the contrastively fine-tuned VLM2Vec model significantly degrades performance (-7.4 points). This suggests that inference-time prompting with generic instructions disrupts the specialized representation space learned for a specific task.

- **Superiority of Training-Time Integration:** Our PDF framework, which integrates diversity-inducing mechanisms during training, yields a substantially more powerful model, outperforming the MetaEoL-prompted baseline by a large margin (+11.6 points overall). This demonstrates the advantage of building robust representations from the ground up over attempting to elicit them post-hoc.

- **Efficiency Advantage:** Our method maintains a 1x inference cost by using a single path, whereas prompt-based ensembling incurs a significant 8x overhead.

In summary, these results validate that our PDF framework's performance gains are attributable to its specific training-time methodology, which is fundamentally more effective and efficient for this task than inference-time, multi-prompt ensembling.

Table 7: **Performance with MetaEoL.** Performance and efficiency comparison with the MetaEoL method. Our PDF framework shows superior performance and efficiency.

| Model | IND | OOD | Overall | Inference Cost |
|---|---|---|---|---|
| VLM2Vec-2B (Baseline) | 65.6 | 52.3 | 59.7 | 1x |
| + MetaEoL | 53.2 | 51.2 | 52.3 | 8x |
| **PDF-VLM2Vec-2B (Ours)** | **69.5** | **56.9** | **63.9** | 1x |
| + MetaEoL | 64.4 | 58.4 | 61.7 | 8x |

## A.5 HYPERPARAMETER EXPERIMENT ON MMEB.

Our PDF framework is governed by three primary hyperparameters: the number of parallel paths $(N)$, the prefix length $(K)$, and the MIM loss weight $(\lambda_{MIM})$. To efficiently determine an ideal configuration, we conduct a sensitivity analysis on the PDF-VLM2Vec-Qwen2VL (2B) model in a low-resolution setting on MMEB. Our search is centered around the configuration of $N = 2$, $K = 20$, and $\lambda_{MIM} = 1 \times 10^{-4}$, while other hyperparameters (*e.g.*, learning rate) are kept consistent with our main experimental setup. The detailed results are presented in Table 8.

As shown in Table 8, the configuration of $N = 2$, $K = 20$, and $\lambda_{MIM} = 1 \times 10^{-4}$ achieves the best overall performance. Notably, while the $N = 4$ setting yields a slightly higher in-distribution (IND) score, it does so at the cost of doubling the training computation. Given this compelling trade-off between performance and efficiency, we adopt $N = 2$ for all main experiments. This choice is further validated by our main results, which demonstrate that the hyperparameters selected in this simplified setting (low-resolution, 2B model) generalize effectively to higher-resolution inputs and the larger 7B model.

Regarding the number of paths, as shown in table A.6, performance does not scale linearly. While $N = 4$ offers slight gains in IND tasks, $N = 8$ degrades performance. The degradation in performance at $N = 8$ corresponds to a failure in maintaining path diversity. As visualized in Figure 4 (b), the average cosine similarity between paths for the $N = 8$ model is the highest, indicating less effective decoupling and a tendency towards representation collapse. In contrast, the better-performing $N = 2$ and $N = 4$ models successfully maintain lower path similarity. We attribute this to the combinatorial explosion of path pairs ($C(N, 2)$) in the MIM objective. For $N = 8$, the model must de-correlate 28 pairs, making the minimax game significantly harder to stabilize compared to $N = 4$ (6 pairs) or $N = 2$ (1 pair).

We also compared our standard Deep Injection against a Shallow Injection variant where prefixes are only injected at the first layer. As shown in Table A.6, the results confirm that Deep Injection is essential for effectively steering the model's internal representations and achieving optimal performance.

Table 8: **Hyperparameter Analysis on MMEB.** Ablation study on the number of parallel paths ($N$), prefix length ($K$), and MIM loss weight ($\lambda_{\text{MIM}}$) using the PDF-VLM2Vec-Qwen2VL-LR model. The results show that the configuration of $N = 2$, $K = 20$, and $\lambda_{\text{MIM}} = 10^{-4}$ achieves the optimal trade-off between performance and efficiency.

| Hyperparameter | Value | Average Score | | |
|---|---|---|---|---|
| | | IND | OOD | Overall |
| $N$ | **2** | 62.8 | **53.3** | **58.6** |
| | 4 | **63.0** | 53.0 | **58.6** |
| | 8 | 62.4 | 53.0 | 58.2 |
| $K$ | 10 | 62.4 | 51.3 | 57.4 |
| | **20** | **62.8** | **53.3** | **58.6** |
| | 40 | 61.9 | 52.3 | 57.7 |
| $\lambda_{\text{MIM}}$ | $1 \times 10^{-3}$ | 62.5 | 50.7 | 57.3 |
| | $\mathbf{1 \times 10^{-4}}$ | **62.8** | **53.3** | **58.6** |
| | $1 \times 10^{-5}$ | 61.9 | 50.1 | 56.7 |
| Injection Method | Shallow | 59.5 | 47.3 | 54.0 |
| | Deep | **62.8** | **53.3** | **58.6** |

## A.6 DETAILS OF MI ESTIMATOR AND AGGREGATION MODULE

**MI Estimator.** Our Mutual Information (MI) estimator is designed to approximate the conditional distribution $q(y|x)$ as a diagonal Gaussian, $\mathcal{N}(\mu, \Sigma)$. To achieve this, it employs two separate Multi-Layer Perceptrons (MLPs) to predict the mean ($\mu$) and the log-variance ($\log \sigma^2$, the diagonal of $\Sigma$).

- **Mean Prediction Network ($\mu$):** This network takes an input of dimension $d_x$, projects it to an intermediate hidden dimension of $d_h/2$, applies a ReLU activation, and then projects it back to the output dimension $d_y$. Its structure is: `Linear(`$d_x$`, `$d_h/2$`)` $\to$ `ReLU` $\to$ `Linear(`$d_h/2$`, `$d_y$`)`.

- **Log-Variance Prediction Network ($\log \sigma^2$):** This network shares an identical architecture with the mean prediction network but includes an additional Tanh activation function at the end. This final Tanh layer serves to constrain the output values, enhancing training stability. Its structure is: `Linear(`$d_x$`, `$d_h/2$`)` $\to$ `ReLU` $\to$ `Linear(`$d_h/2$`, `$d_y$`)` $\to$ `Tanh`.

In our experiments, the input/output dimensions ($d_x, d_y$) are equal to the model's hidden size $d$, and the intermediate hidden dimension $d_h$ is set to $4d$.

**Aggregation Module.** The aggregation module is a lightweight network designed to compute the fusion weights for the $N$ parallel embeddings. It first concatenates the $N$ embeddings, each of dimension $d$, into a single vector of size $N \times d$. This vector is then processed by an MLP followed by a Softmax layer.

- **MLP:** The MLP consists of two linear layers with a Swish (SiLU) activation function in between. It maps the concatenated input vector from $N \times d$ to an intermediate dimension $d$, and then down to a vector of size $N$. The structure is: `Linear`$(N \times d,\ d) \rightarrow$ `SiLU` $\rightarrow$ `Linear`$(d,\ N)$.

- **Softmax:** A Softmax function is applied to the final $N$-dimensional output of the MLP to produce a set of normalized weights, which are then used for the weighted average of the parallel embeddings.

## A.7   QUALITATIVE RESULTS

To further verify the effectiveness of our PDF framework, we conduct a visual comparison between the VLM2Vec-Qwen2VL (7B) baseline and our PDF-enhanced model on the MMEB dataset.

As illustrated in Fig. 6 and Fig. 7, including various multi-modal retrieval tasks, our PDF-enhanced model demonstrates a superior understanding of complex multi-modal queries, which require the model to retrieve target matching the requirement of image and text contents. As shown in the third row of Fig. 6, the retrieved text of VLM2Vec can correctly matches the image content but neglects the crucial relationship specified in the query text. In contrast, our PDF-VLM2Vec successfully distinguishes this subtle difference and retrieves the correct result.

We attribute this improvement to our PDF, which compels the model to explore multifaceted features. Hence, our model trained with PDF can distinguish targets based on a wider range of aspects, leading to more precise and robust retrieval.

## A.8   DETAIL RESULTS OF THE BASELINE AND OUR VLM2VEC ON MMEB

We present the detailed results of each model on various datasets of MMEB in Table 9.

## A.9   LIMITATIONS AND FUTURE WORK

While our PDF framework demonstrates significant improvements, we acknowledge several limitations that also point to promising directions for future research. Our ablation studies (see Appendix A.5) show that performance does not monotonically increase with the number of paths, degrading when scaling from N=4 to N=8. We attribute this to the quadratic growth of path pairs ($C(N, 2)$) that the MIM loss must de-correlate, which may lead to optimization challenges. Future work could explore more scalable or hierarchical MIM objectives to support a larger number of parallel paths.

## A.10   THE USE OF LARGE LANGUAGE MODELS (LLMs)

Large Language Models (LLMs) played a significant role as a writing assistant throughout the composition of this manuscript. Specifically, we utilized LLMs for tasks such as paraphrasing, grammar correction, and improving the clarity and conciseness of the text. The core scientific ideas, experimental design, data analysis, and conclusions presented in this paper were conceived and formulated entirely by the human authors. The LLMs' role was strictly limited to refining the linguistic expression of these pre-existing ideas. This transparent disclosure aligns with our commitment to academic integrity and responsible research practices.

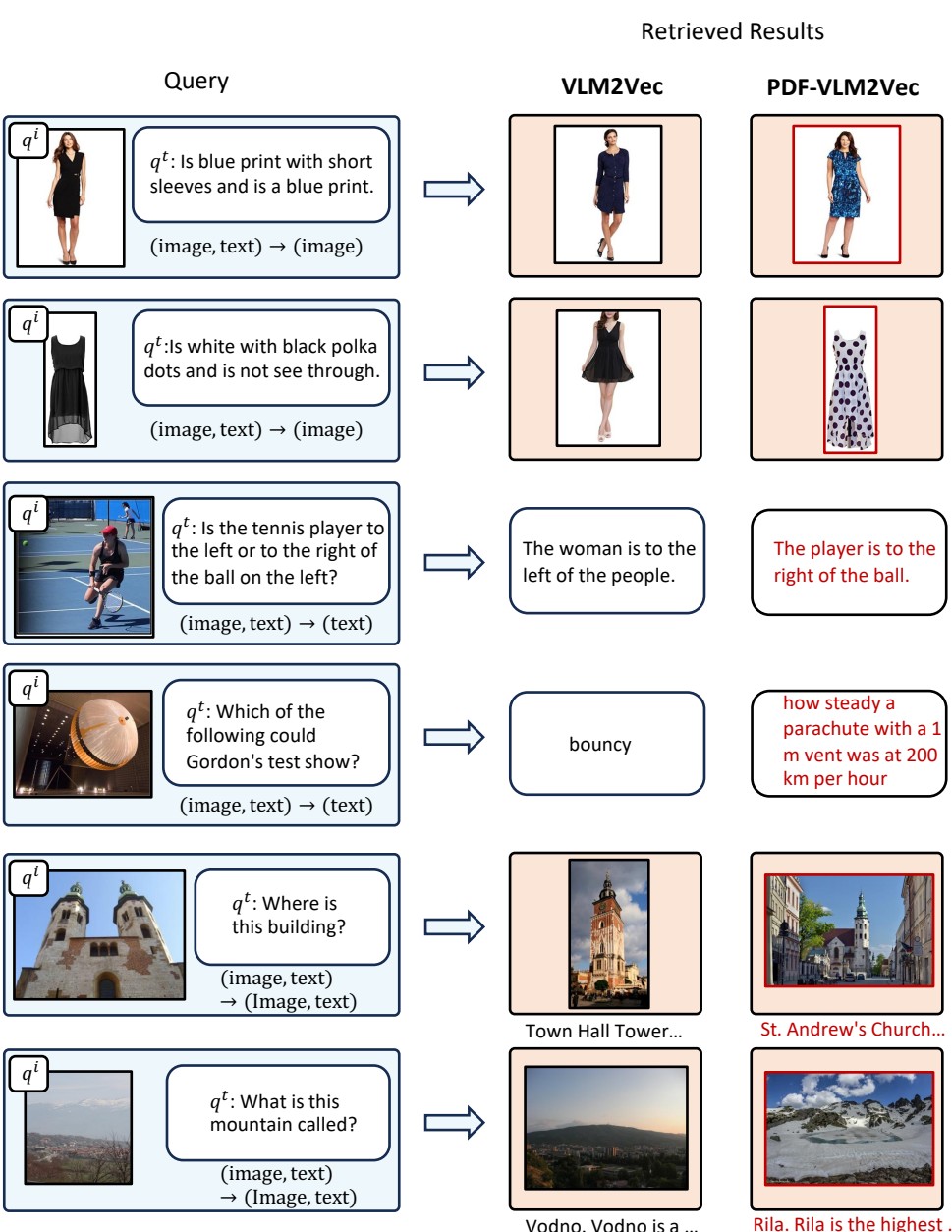

Figure 6: **Qualitative Results Part 1.** We show the results of our method across six different retrieval tasks compared with VLM2Vec-Qwen2VL-7B.

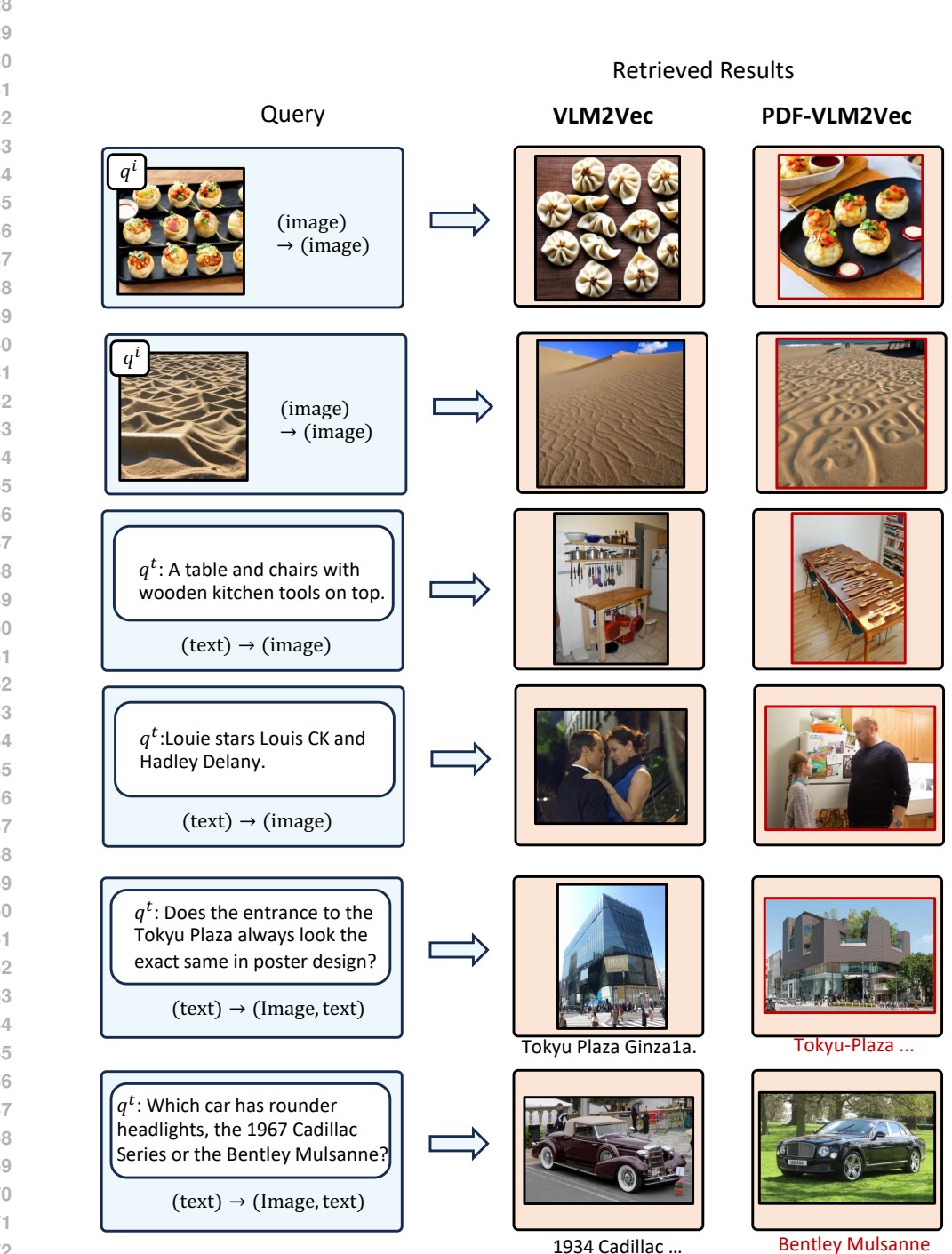

Figure 7: **Qualitative Results Part 2.** We show the results of our method across six different retrieval tasks compared with VLM2Vec-Qwen2VL-7B.

Table 9: **Detailed Results on MMEB.** Detailed per-task performance comparison on the MMEB benchmark across all 36 tasks. The table reports Precision@1 for each task. Tasks marked with a highlighted background are from the out-of-distribution (OOD) test sets. **L** and **Q** means the LLAVA-1.6 and Qwen2VL backbones respectively.

| Task | CLIP | OpenCLIP | SigLIP | BLIP2 | UniIR | VLM2Vec | | | PDF-VLM2Vec | | |
|------|------|----------|--------|-------|-------|---------|---|---|-------------|---|---|
| | | | | | | L-7B | Q-2B | Q-7B | L-7B | Q-2B | Q-7B |
| **Classification (10 tasks)** | | | | | | | | | | | |
| ImageNet-1K | 55.8 | 63.5 | 45.4 | 10.3 | 58.3 | 74.5 | 77.2 | 81.2 | 73.2 | 79.4 | 80.3 |
| N24News | 34.7 | 38.6 | 13.9 | 36.0 | 42.5 | 80.3 | 76.2 | 79.4 | 80.1 | 78.6 | 82.5 |
| HatefulMemes | 51.1 | 51.7 | 47.2 | 49.6 | 56.4 | 67.9 | 61.5 | 67.7 | 64.6 | 64.6 | 70.3 |
| VOC2007 | 50.7 | 52.4 | 64.3 | 52.1 | 66.2 | 91.5 | 79.0 | 81.7 | 89.6 | 82.7 | 84.3 |
| SUN397 | 43.4 | 68.8 | 39.6 | 34.5 | 63.2 | 75.8 | 73.9 | 79.2 | 76.4 | 75.0 | 78.9 |
| Place365 | 28.5 | 37.8 | 20.0 | 21.5 | 36.5 | 44.0 | 36.0 | 38.4 | 43.4 | 41.4 | 42.0 |
| ImageNet-A | 25.5 | 14.2 | 42.6 | 3.2 | 9.8 | 43.6 | 51.5 | 55.3 | 43.0 | 53.8 | 53.9 |
| ImageNet-R | 75.6 | 83.0 | 75.0 | 39.7 | 66.2 | 79.8 | 86.4 | 74.5 | 77.9 | 89.2 | 85.7 |
| ObjectNet | 43.4 | 51.4 | 40.3 | 20.6 | 32.2 | 39.6 | 22.5 | 38.2 | 35.8 | 28.6 | 45.8 |
| Country-211 | 19.2 | 16.8 | 14.2 | 2.5 | 11.3 | 14.7 | 22.3 | 31.0 | 12.9 | 27.2 | 30.3 |
| *All Classification* | 42.8 | 47.8 | 40.3 | 27.0 | 44.3 | 61.2 | 58.7 | 62.7 | 59.7 | 62.1 | 65.4 |
| **VQA (10 tasks)** | | | | | | | | | | | |
| OK-VQA | 7.5 | 11.5 | 2.4 | 8.7 | 25.4 | 69.0 | 48.1 | 57.2 | 70.3 | 58.4 | 67.4 |
| A-OKVQA | 3.8 | 3.3 | 1.5 | 3.2 | 8.8 | 54.4 | 40.3 | 48.0 | 58.0 | 51.2 | 59.1 |
| DocVQA | 4.0 | 5.3 | 4.2 | 2.6 | 6.2 | 52.0 | 85.2 | 90.0 | 80.4 | 89.5 | 92.4 |
| InfographicsVQA | 4.6 | 4.6 | 2.7 | 2.0 | 4.6 | 30.7 | 49.3 | 65.0 | 42.0 | 56.5 | 68.1 |
| ChartQA | 1.4 | 1.5 | 3.0 | 0.5 | 1.6 | 34.8 | 42.0 | 55.3 | 45.7 | 50.0 | 60.2 |
| Visual7W | 4.0 | 2.6 | 1.2 | 1.3 | 14.5 | 49.8 | 50.1 | 53.0 | 52.5 | 52.6 | 54.6 |
| ScienceQA | 9.4 | 10.2 | 7.9 | 6.8 | 12.8 | 42.1 | 29.2 | 39.5 | 40.5 | 36.4 | 43.0 |
| VizWiz | 8.2 | 6.6 | 2.3 | 4.0 | 24.3 | 43.0 | 37.0 | 38.5 | 45.1 | 43.8 | 46.6 |
| GQA | 41.3 | 52.5 | 57.5 | 9.7 | 48.8 | 61.2 | 47.9 | 52.7 | 53.3 | 42.0 | 56.4 |
| TextVQA | 7.0 | 10.9 | 1.0 | 3.3 | 15.1 | 62.0 | 63.7 | 70.2 | 73.6 | 73.7 | 81.7 |
| *All VQA* | 9.1 | 10.9 | 8.4 | 4.2 | 16.2 | 49.9 | 49.3 | 56.9 | 56.1 | 55.4 | 63.0 |
| **Retrieval (12 tasks)** | | | | | | | | | | | |
| VisDial | 30.7 | 25.4 | 21.5 | 18.0 | 42.2 | 80.9 | 75.5 | 81.3 | 82.7 | 80.0 | 82.7 |
| CIRR | 12.6 | 15.4 | 15.1 | 9.8 | 51.3 | 49.9 | 48.5 | 50.0 | 51.8 | 51.6 | 53.0 |
| VisualNews_t2i | 78.9 | 74.0 | 51.0 | 48.1 | 74.3 | 75.4 | 74.5 | 80.2 | 74.9 | 74.3 | 80.0 |
| VisualNews_i2t | 79.6 | 78.0 | 52.4 | 13.5 | 76.8 | 80.0 | 74.5 | 82.4 | 78.0 | 76.6 | 82.4 |
| MSCOCO_t2i | 59.5 | 63.6 | 58.3 | 53.7 | 68.5 | 75.7 | 71.2 | 77.2 | 76.9 | 73.1 | 76.9 |
| MSCOCO_i2t | 57.7 | 62.1 | 55.0 | 20.3 | 72.1 | 73.1 | 68.2 | 73.2 | 72.7 | 69.6 | 73.8 |
| NIGHTS | 60.4 | 66.1 | 62.9 | 56.5 | 66.2 | 65.5 | 65.1 | 67.9 | 67.2 | 68.0 | 68.1 |
| WebQA | 67.5 | 62.1 | 58.1 | 55.4 | 89.6 | 87.6 | 86.1 | 88.1 | 89.1 | 87.7 | 87.8 |
| FashionIQ | 11.4 | 13.8 | 20.1 | 9.3 | 40.2 | 16.2 | 13.5 | 16.8 | 16.1 | 15.7 | 16.8 |
| Wiki-SS-NQ | 55.0 | 44.6 | 55.1 | 28.7 | 12.2 | 60.2 | 57.7 | 61.4 | 67.0 | 58.6 | 65.6 |
| OVEN | 41.1 | 45.0 | 56.0 | 39.5 | 69.4 | 56.5 | 64.5 | 67.4 | 49.1 | 68.3 | 70.8 |
| EDIS | 81.0 | 77.5 | 23.6 | 54.4 | 79.2 | 87.8 | 80.1 | 87.1 | 88.5 | 81.2 | 81.8 |
| *All Retrieval* | 53.0 | 52.3 | 31.6 | 33.9 | 61.8 | 67.4 | 65.0 | 69.4 | 67.8 | 67.1 | 70.0 |
| **Visual Grounding (4 tasks)** | | | | | | | | | | | |
| MSCOCO | 33.8 | 34.5 | 46.4 | 28.9 | 46.6 | 80.6 | 66.3 | 79.1 | 82.4 | 71.2 | 78.1 |
| RefCOCO | 56.9 | 54.2 | 70.8 | 47.4 | 67.8 | 88.7 | 80.8 | 87.4 | 93.3 | 85.4 | 91.1 |
| RefCOCO-matching | 61.3 | 68.3 | 50.8 | 59.5 | 62.9 | 84.0 | 74.4 | 83.1 | 89.1 | 84.6 | 91.1 |
| Visual7W-pointing | 55.1 | 56.3 | 70.1 | 52.0 | 71.3 | 90.9 | 70.0 | 79.9 | 91.8 | 79.2 | 85.8 |
| *All Visual Grounding* | 51.8 | 53.3 | 59.5 | 47.0 | 65.3 | 86.1 | 72.9 | 82.2 | 89.2 | 80.3 | 86.5 |
| **Final Score (36 tasks)** | | | | | | | | | | | |
| All | 37.8 | 39.7 | 34.8 | 25.2 | 44.7 | 62.9 | 59.7 | 65.5 | 64.7 | 63.9 | 68.6 |
| All IND | 37.1 | 39.3 | 32.3 | 25.3 | 47.1 | 67.5 | 65.6 | 71.9 | 70.4 | 69.5 | 74.0 |
| All OOD | 38.7 | 40.2 | 38.0 | 25.1 | 41.7 | 57.1 | 52.3 | 57.5 | 57.5 | 56.9 | 61.8 |

