# OpenReview forum: "Explore More, Learn Better: Parallel MLLM Embeddings under Mutual Information Minimization"
_ICLR.cc/2026/Conference — Submitted to ICLR 2026_

### Official Review · Reviewer_9b5b · 2025-10-22

**Soundness:** 3
**Presentation:** 3
**Contribution:** 3
**Rating:** 6
**Confidence:** 3

**Summary:**

The paper introduces a novel Parallel Decoupling Framework (PDF) designed to improve the training of MLLMs for embedding learning. The main idea is to exploit the steerability of MLLMs by generating multiple parallel embeddings for a single input. This approach is realized by conditioning the MLLM backbone on distinct, learnable prefixes, which in turn create multiple parallel paths for embedding generation. Here is the core contributions of the paper:

1) Parallel Embedding Generation: The paper proposes using a shared MLLM backbone with multiple parallel paths, each conditioned by unique prefixes. This allows the model to generate diverse embeddings, fully utilizing the input's semantic richness.

2) Mutual Information Minimization (MIM): To avoid collapse into redundant embeddings, the paper introduces a mutual information minimization objective. This explicitly minimizes statistical dependencies between parallel embeddings, enhancing diversity and ensuring robust semantic coverage.

3) Training Efficiency: The framework introduces minimal computational overhead during inference, as the system can select a single path for inference, maintaining the benefits of parallel training without extra cost.

4) Empirical Validation: The PDF framework is instantiated on several MLLM backbones, such as VLM2Vec with LLaVA and Qwen2VL, achieving significant performance improvements across multiple benchmarks (e.g., MMEB), with gains ranging from +4.2% to +8.9% over baseline models. The model also shows superior training efficiency, surpassing the baseline in fewer iterations and with reduced computational budget.

**Strengths:**

The paper’s strengths span multiple dimensions. In terms of originality, it presents a fresh reformulation of multimodal embedding learning by replacing the conventional Single input–Singular embedding–Contrastive supervision (SSC) paradigm with the proposed Single input–Parallel paths–Parallel outputs (SPP) paradigm. This idea, realizing multiple, decorrelated embeddings via deep prefix injection and mutual information minimization, creatively leverages the inherent steerability of MLLMs, representing a novel and well-motivated contribution. In terms of quality, the technical design is rigorous: the dual-objective optimization (contrastive loss + variational MIM) is theoretically grounded, experimentally validated across multiple MLLM backbones, and supported by comprehensive ablation, efficiency, and zero-shot evaluations. In addition, the paper is well organized, with clear figures and consistent notations that make complex mechanisms (e.g., prefix-based parallel paths and MI estimation) easy to follow. Regarding significance, the proposed framework achieves notable gains (up to +8.9% on MMEB) while requiring negligible inference overhead, suggesting strong practical impact and potential influence on future MLLM embedding research. Overall, the work is a empirically convincing contribution to representation learning for multimodal foundation models

**Weaknesses:**

1. Most experiments are conducted on the MMEB benchmark, with limited evaluation across other multimodal embedding datasets (e.g., MSCOCO, LAION-Aesthetics, or ImageNet-Text retrieval). This restricts the evidence for generalization and may introduce benchmark-specific bias. Adding results on diverse domains would better demonstrate robustness and transferability.

2. The paper proposes learnable prefixes for parallel embedding generation but provides little insight into how prefix dimensionality, initialization, or number of paths 𝑁 affect performance. A systematic analysis (e.g., scaling 𝑁=2, 4, 8) could clarify whether improvements stem from architectural diversity or training regularization.

3. The framework is demonstrated for multimodal retrieval and classification, but its applicability to downstream generation or reasoning tasks is not discussed. Since MLLMs are often used in generative settings, evaluating or at least discussing potential extensions to instruction-following or captioning would broaden the paper’s impact.

4. The baselines largely include VLM2Vec and a few classic models (e.g., CLIP, BLIP2). However, more recent multimodal embedding approaches (e.g., MM-VLM, EVA-CLIP, E5-V, MME5) are not included or discussed in depth. A more thorough comparative evaluation would improve contextual positioning and claim of novelty.

5. Every method has limitations. For example, perhaps in data-scarce settings, adding many paths could over­fit; or perhaps for very simple inputs, parallel embeddings may be redundant. The paper would improve by including a short discussion of when the proposed method might not offer gains (and perhaps empirical evidence of such cases).

**Questions:**

Please see weaknesses.

---

> ### Author Response · Authors · 2025-11-23
> **A). Generalization beyond MMEB**
>
> We appreciate the suggestion to further explore generalization, particularly in long-form and RAG-like scenarios. We demonstrate the superiority of PDF through three types of evidence:
>
> **1. Strong Generalization on MMEB OOD Split:**
>
> We chose MMEB as the primary benchmark specifically because it includes a dedicated Out-Of-Distribution (OOD) split containing 16 datasets not seen during training. As shown in Table 1, our method achieves significant gains on this OOD split (+4.6% for 2B, +4.3% for 7B), indicating strong capability to generalize to novel/unseen domains.
>
> **2. Long-Form Retrieval & RAG Capabilities (Zero-Shot):**
>
> To address your concern about long-form retrieval (a proxy for RAG performance), we point to our zero-shot evaluation on ShareGPT4V and Urban1K benchmarks in Section 4.2 (Line323, Table 2). These benchmarks are particularly relevant as they are characterized by highly detailed, long-form captions that simulate the complex documents typically found in RAG systems.
>
> Performance: Our PDF-VLM2Vec-2B achieves a massive +13.2% improvement on Urban1K and +5.3% on ShareGPT4V compared to the baseline. The 7B model also shows consistent gains (+2.9%/+2.3%). This confirms that our "diverse exploration" paths significantly aid in capturing the nuanced semantics required for long-context retrieval.
>
> Latency-Quality Trade-off: Regarding the RAG latency concern, since our inference uses a single path (detailed in Table 4), we achieve performance gains with zero additional latency compared to the standard VLM2Vec, offering an optimal Pareto frontier for RAG systems.
>
> **3. Additional Experiments on Unseen Datasets:**
>
> To further validate generalization beyond MMEB, we conducted new experiments on Fashion200K (e-commerce domain) and InfoSeek (information seeking). As shown in the table below, our PDF-VLM2Vec consistently outperforms the VLM2Vec baseline across different model scales and query modalities (Image-Text, Text-Image, etc.).
>
> | Model             | Image-Text  | Image+Text-Text | Text-Image  | Image+Text-Image+Text |
> |:------------------|:-----------:|:---------------:|:-----------:|:---------------------:|
> |                   | Fashion200K | InfoSeek        | Fashion200K | InfoSeek              |
> |                   | (R@10)      | (R@5)           | (R@10)      | (R@5)                 |
> | VLM2Vec-Qwen2VL-2B        | 4.7         | 30.4            | 5.2         | 41.5                  |
> | PDF-VLM2Vec-Qwen2VL-2B    | 7.7         | 33.9            | 12.7        | 41.4                  |
> |---|---|---|---|---|
> | VLM2Vec-Qwen2VL-7B        | 9.6         | 37.2            | 13.6        | 44.7                  |
> | PDF-VLM2Vec-Qwen2VL-7B    | 10.2        | 42.1            | 14.1        | 44.6                  |

---

> ### Author Response · Authors · 2025-11-23
> **B). On Hyperparameter Analysis**
>
> We thank the reviewer for highlighting a more systematic analysis of hyperparameters. We address the concerns regarding the number of paths (N), prefix dimensionality (d), and initialization below.
>
> **1. Analysis on the Number of Paths (N)**
>
> | Path Number | IND  | OOD  | ALL  |
> |:-----------:|:----:|:----:|:----:|
> |      2      | 62.8 | 53.3 | 58.6 |
> |      4      | 63.0 | 53.0 | 58.6 |
> |      8      | 62.4 | 53.0 | 58.2 |
>
>
> This is the most critical hyperparameter influencing the architectural diversity. In our original submission (Appendix A.1), we provided an initial study. Here, we extended our study to include N=8. As shown in the table above, performance does not scale linearly with the number of paths. While N=4 offers slight gains on IND tasks, N=8 degrades performance. A deeper analysis reveals the underlying reasons for this result:
>
> - **Combinatorial Explosion in the MIM Training Objective**: The number of pairwise comparisons required by our MIM loss grows quadratically with the number of paths, N, following the function N(N−1)/2. For N=2/4, this results in just 1/6 pairs, while for N=8, there are 28 pairs that must be optimized. This dramatically increases the complexity of the two-stage adversarial-like optimization, making it substantially more difficult to find a stable equilibrium that balances contrastive leanring with inter-path diversity.
> - **Consequent Representation Collapse**: As N increases, the optimization becomes challenging, leading the model to a "lazy" solution where paths become similar. This phenomenon is empirically validated by our analysis of the average cosine similarity between paths, as presented in the Figure 4 (a) in the appendix, that is, higher similarity for N=8. The model fails to learn diverse representations, making additional paths redundant.
> - **Suboptimal Fusion of Redundant Paths**: As N increases, path aggregation will be overwhelmed by a large number of similar representations. This may result in attention weights being diluted by redundant information, making it difficult to focus on the useful few paths, ultimately reducing the quality of final results.
>
> This above analysis clarifies that the improvements stem from a "sweet spot" of architectural diversity (N=2 or 4) combined with effective training regularization from the MIM objective. Too much diversity (N=8) hinders optimization, leading to performance degradation.
>
> **2. Prefix Dimensionality and Initialization:**
>
> - Prefix Dimensionality (d): We set the dimension d of our prefix vectors to be identical to the hidden dimension of the LLM's transformer layers. This is a standard and necessary choice for our deep injection mechanism, which directly concatenates the prefix's keys and values with the input's keys and values (Eq. (2)). While one could explore projecting prefixes from a lower-dimensional space, we opted for this direct approach for simplicity and maximum expressive power. Our initial study on prefix length K (the number of prefix tokens) in Appendix Table 8 showed that K=20 provided a good balance.
> For the other hyperparameters, we followed standard practices as they were not the primary focus of our methodological innovation.
> - Initialization: We initialize the learnable prefix parameters using a standard normal distribution (mean=0, std=0.02). This is a common practice for initializing new parameters in neural networks to ensure stable training start. Since our framework proved robust with this standard initialization, we did not explore more complex initialization schemes.
>
> In summary, our detailed analysis on N confirms that the performance gains are a result of well-calibrated architectural diversity and regularization. For other parameters like dimensionality and initialization, we adopted standard, well-motivated practices. We will ensure this comprehensive discussion is included in the revised appendix.

---

> ### Author Response · Authors · 2025-11-23
> **C). Extending to Generation and Reasoning Tasks**
>
> This is an excellent question that touches upon the broader potential of our framework. We agree that extending the concept of parallel, de-correlated paths to generative and reasoning tasks is a fascinating and promising direction.
>
> Our current work is deliberately focused on establishing the effectiveness of the PDF framework for the fundamental task of universal multimodal embedding learning. As the reviewer notes, high-quality embeddings are themselves a cornerstone that can empower a wide range of downstream applications, most notably in Retrieval-Augmented Generation (RAG) systems where our improved embeddings can lead to more relevant and accurate document retrieval.
>
> That said, the core principle of PDF—generating diverse outputs via parallel paths under an explicit MIM constraint—is highly generalizable. As you suggested, we envision an exciting application in generation and reasoning:
> - For Generation and Reasoning: In the reasoning task, different parallel paths could be trained to explore different "thought" trajectories. By encouraging these paths to be different, the model is less likely to get stuck in a single, potentially flawed, line of reasoning.
>
> These extensions represent significant research directions that warrant dedicated investigation. We believe our work lays a solid foundation for such future explorations. We thank the reviewer for this forward-thinking suggestion and will add a discussion of these potential future directions to the conclusion of our revised paper to broaden its impact. Due to constraints in training costs and training data, we will explore this in the future work.

---

> ### Author Response · Authors · 2025-11-23
> **D). Comparisons with More Models**
>
> We thank the reviewer for suggesting a comparison with more recent multimodal embedding models. To better contextualize the work, we have compiled a table comparing our 7B model with these embedding models, using their reported scores on the MMEB benchmark.
>
> | Model                                           | Classification | VQA   | Retrieval | Grounding | IND   | OOD   | Overall |
> |:------------------------------------------------|:--------------:|:-----:|:---------:|:---------:|:-----:|:-----:|:-------:|
> | EVA-CLIP-8B                                     | 56.0           | 10.4  | 49.2      | 58.9      | 38.1  | 45.6  | 43.7    |
> | E5-V                                            | 21.8           | 4.9   | 11.5      | 19.0      | 14.9  | 11.5  | 13.3    |
> | MMRet-MLLM-7B                                   | 47.2           | 18.4  | 56.5      | 62.2      | -     | -     | 44.0    |
> | MMRet-MLLM-7B (ft)                              | 56.0           | 57.4  | 69.9      | 83.6      | -     | -     | 64.1    |
> | mmE5-7B (w/ 560K synthetic data + labeled data) | 67.6           | 62.8  | 70.9      | 89.7      | 72.3  | 66.7  | 69.8    |
> | PDF-VLM2Vec (7B)                                | 65.4           | 63.0  | 70.0      | 86.5      | 74.0  | 61.8  | 68.6    |
>
>
> This comparison highlights several key points:
>
> - *Superior Performance on In-Distribution Data*: It is crucial to note that mmE5-7B's performance relies on 560K additional high-quality synthetic data, which were specifically generated to cover a wide range of tasks. Even without access to this extra data, our model outperforms mmE5-7B on the in-distribution (IND) portion of the benchmark by +1.7 points (74.0 vs. 72.3). This shows the remarkable effectiveness of our training framework in maximizing performance from the given training data. The superior OOD performance of mmE5 is likely attributable to its synthetic data covering aspects of the OOD test sets.
> - *Orthogonal Contribution*: Most importantly, our work is orthogonal to data-centric approaches like mmE5 and MMRet (which also leverages large-scale synthesized data). They focus on improving performance by curating better training data, while our PDF framework focuses on innovating the training paradigm itself. These two directions are complementary and could likely be combined for even greater performance gains (e.g., by applying our PDF training on top of a model pre-trained with synthetic data).
>
> In summary, our method achieves SOTA-level performance through a novel training framework alone, distinguishing its contribution from data-centric methods and highlighting its potential for broader applicability.

---

> ### Author Response · Authors · 2025-11-23
> **E). Limitations of the Proposed Method**
>
> We thank the reviewer for this thoughtful suggestion. We agree that a discussion of limitations is essential and will add a dedicated "Limitations and Future Work" section to the paper. Our primary limitations are:
>
> - Scalability of MIM with Path Numbers (Internal Limitation): As our own analysis for W2 showed, the performance of our framework does not monotonically increase with the number of paths N. We observed performance degradation when scaling from N=4 to N=8. We attribute this to the quadratic growth in the number of path pairs that the MIM loss needs to de-correlate (C(N,2)). This suggests that our current MIM implementation may face optimization challenges when N becomes very large, presenting a scalability limit. Future work could explore more scalable or hierarchical MIM objectives to address this.
>
> - Data Scarcity (External Limitation): As the reviewer rightly pointed out, our method introduces additional learnable parameters (the deep prefixes). In data-scarce settings, training multiple parallel paths could potentially increase the risk of overfitting. Our framework is likely to be most effective when sufficient training data is available to support the learning of diverse, meaningful representations.
>
> We believe that transparently discussing these limitations will provide a more nuanced understanding of our method's applicability and pave the way for future improvements. We are grateful for the reviewer's suggestion to include this.

---

> > ### Comment · Reviewer_9b5b · 2025-11-26
> >
> > Thank you for the detailed response from the authors. My concern has been almost addressed. For now, I keep my current score and will make a final decision after considering the other reviewers’ comments.

---

> > > ### Author Response · Authors · 2025-11-26
> > >
> > > We are particularly encouraged that our rebuttal successfully addressed your previous concerns. Your insightful comments have been invaluable in improving our paper, and we truly appreciate your support.

---

### Official Review · Reviewer_A3ti · 2025-10-26

**Soundness:** 2
**Presentation:** 2
**Contribution:** 2
**Rating:** 4
**Confidence:** 4

**Summary:**

The key idea is to replace the standard “single input–single embedding” paradigm with a deep prefix injection mechanism that conditions multiple parallel paths through learnable prefix vectors injected at each transformer layer.

**Strengths:**

1. The deep prefix injection design is a clever extension of prefix-tuning that operates at every layer, rather than only at the input, offering a plausible mechanism to steer MLLMs toward richer, more diverse embedding spaces.
2. The authors present extensive benchmarks on multiple model scales and backbones

**Weaknesses:**

1. Lack of evidence that deep prefix injection truly induces semantic diversity. There is no analysis of attention distributions, token contributions, or feature subspace diversity.
2. Table 3 shows nearly identical performance between Single Prefix and Aggregate inference strategies. If the learned embeddings are genuinely diverse, aggregation should bring at least a small improvement. This raises reasonable doubt that the parallel paths have collapsed or that diversity is not reflected in performance.
3. No demonstration that deep injection outperforms prompt-based methods. MetaEoL [1] is relevant baseline. Without a direct comparison, it is unclear whether the reported improvements arise from deep injection itself or from generic multi-prompt effects. Moreover, prompt-based methods usually offer better interpretability—something the current approach lacks.

[1] Meta-Task Prompting Elicits Embeddings from Large Language Models

**Questions:**

1. The scores in Table 3 are almost identical. Is this expected? Please report the individual performance of each path to clarify whether the embeddings are genuinely diverse. If diversity exists, why does aggregation bring no additional benefit?
2. To substantiate the “different subspace / computation trajectory” claim, please include stronger evidence.
3. Please provide quantitative comparison with MetaEoL or similar prefix/prompting methods under the same setup. This would clarify whether “deep injection” offers real gains beyond prompting and whether the improvements come from deeper modulation or simply from prompt-like effects.

---

> ### Author Response · Authors · 2025-11-23
> **A). Clarification of Our Main Contribution**
>
> We thank the reviewer for the constructive feedback and for acknowledging the novelty of our deep prefix injection design.
>
> We would like to begin by clarifying that our core contribution is the Parallel Decoupling Framework (PDF) itself, a holistic training paradigm. This framework comprises three integral components: (1) a parallel-path architecture instantiated by deep prefix injection; (2) an explicit diversity objective using Mutual Information Minimization (MIM); (3) a dual-objective training (Contrastive Alignment and Mutual Information Minimization) to balance representation quality and path diversity. The performance gains are driven by the effective synergy between these components, not just by deep prefix injection alone.
>
> With this clarification, we now address the specific weaknesses raised in review.

---

> ### Author Response · Authors · 2025-11-23
> **B). "Different Subspace/Computation Trajectory" Claim and Aggregation Performance**
>
> Thank you to the reviewer for pushing us to dissect and clarify the diversity learned by our PDF framework. The reviewer's concern is predicated on a strong underlying assumption: that achieving high-quality representations necessitates a two-stage process. First, one must use highly distinct (or even orthogonal) prefix inputs to compel diverse outputs from the model's multiple paths. Subsequently, these diverse outputs must be aggregated at inference time to yield the final, superior representation.
>
> However, our empirical findings point to a more nuanced reality, challenging this assumption. Specifically, the optimal solution is not to maximize inter-path dissimilarity, but rather for path diversity to function as a calibrated regularizer that promotes meaningful semantic contrast. In other words, a subtle yet unique form of diversity is what proves to be optimal. This conclusion is empirically corroborated by Appendix Table 6, where performance degrades significantly under both an excessively large MIM weight ($λ_{MIM}$), which enforces maximal inter-path divergence, and a negligible weight, which results in inter-path redundancy. Such empirical findings are key insights of our work.
>
> **1. Clarifying Our Claim: Diversity as a Regularizer**
>
> We wish to clarify that our central claim is not necessarily to learn in orthogonal subspaces and aggregate complementarity during inference. As stated in the main paper (Line 97-98), "This dual objective training  plays a powerful regularization role on the shared model backbone." We use MIM-enforced diversity as a training-time mechanism to force the model to explore the data manifold from slightly different perspectives, thus preventing it from overfitting to a single, narrow view and improving the generalization capabilities of the shared backbone.
>
> **2. Visualizing the Diversity Nature: Similar Semantics, Different Focus**
>
> To provide stronger evidence for this "regularization" view, we visualized the last layer attention maps from the last token for two parallel paths processing the same input sample, as shown in the Appendix Figure 5. This visualization offers critical insights:
> - **Shared Semantic Anchors**: The parallel paths exhibit substantial semantic overlap, both converging on the core concepts required by the primary contrastive objective. They are not learning unrelated representations.
> - **Subtle Attentional Shifts**: The MIM-enforced diversity manifests not as a radical semantic divergence, but as subtle shifts in attentional focus and weighting. This stands in stark contrast to methods like MetaEoL, where explicit, high-level prompts are used to elicit fundamentally different semantic aspects.
>
> **3. Reinterpreting the Aggregation Performance**
>
> This "subtle diversity" brought by regularization naturally explains the performance of aggregation. Since both paths are already high-performing experts capturing the full core semantics, a simple averaging at inference time offers little room for gains. The true benefit of the diversity has already been "baked into" the powerful shared backbone during training.
>
> In conclusion, our framework leverages minor, MIM-induced variations in computational trajectories as a potent regularizer, rather than creating fundamentally disparate paths for aggregation. This unique mechanism is why a single path at inference can inherit the full benefit of the enriched training, achieving state-of-the-art performance with zero extra cost. We will add this visualization and detailed discussion to the appendix to clarify this important point.
>
> We present the performance of each path in the Table below.
>
> | Model       | Classification | VQA   | Retrieval | Grounding | IND   | OOD   | Overall |
> |:------------|:--------------:|:-----:|:---------:|:---------:|:-----:|:-----:|:-------:|
> | path_0      | 62.09          | 55.42 | 67.08     | 80.30     | 69.59 | 56.84 | 63.92   |
> | path_1      | 62.14          | 55.40 | 66.97     | 80.38     | 69.53 | 56.88 | 63.90   |
> | aggregation | 62.05          | 55.41 | 67.06     | 80.28     | 69.53 | 56.86 | 63.90   |

---

> ### Author Response · Authors · 2025-11-23
> **C). Comparisons with Prompt-based Methods like MetaEoL**
>
> We thank the reviewer for excellent suggestions. To clarify whether the gains of our PDF framework come from the proposed training pipeline or from generic multi-prompt effects, we directly compared it with a strong prompt-based baseline following MetaEoL [1], shown in the table below.
>
> - **VLM2Vec-MetaEoL Experimental Setup:** Adhering to the methodology from MetaEoL, a single image-text pair was processed through VLM2Vec eight times. Each pass utilized one of the eight distinct prompts defined in MetaEoL, which collectively span four meta-tasks. The final representation was then computed by averaging the eight resulting output embeddings. We instantiate it with Qwen2VL based models.
>
> | Model                  | Classification | VQA   | Retrieval | Grounding | IND   | OOD   | Overall | Inference Time |
> |:-----------------------|:--------------:|:-----:|:---------:|:---------:|:-----:|:-----:|:-------:|:--------------:|
> | VLM2Vec-2B             | 58.7           | 49.3  | 65.0      | 72.9      | 65.6  | 52.3  | 59.7    | 1x             |
> | VLM2Vec-2B-MetaEoL     | 46.3           | 43.1  | 60.5      | 65.5      | 53.2  | 51.2  | 52.3    | 8x             |
> | PDF-VLM2Vec-2B         | 62.1           | 55.4  | 67.1      | 80.3      | 69.5  | 56.9  | 63.9    | 1x             |
> | PDF-VLM2Vec-2B-MetaEoL | 58.8           | 55.0  | 64.8      | 76.4      | 64.4  | 58.4  | 61.7    | 8x             |
>
> - **Analysis:**
>   1. *Gains are Not from Generic Multi-Prompting*: Applying the MetaEoL multi-prompting idea to VLM2Vec drastically degrades performance by -7.4 points. This demonstrates that for a model already fine-tuned for a specific embedding task, applying generic, unseen prompts during inference may not necessarily improve performance, but may instead disrupt their learning representation space. This also confirms that the significant +4.2 point gains in our PDF framework does not come from multi-prompt effects.
>   2. *Our PDF Framework is Superior to Superficial Multi-Prompt* by a massive +11.6 points (63.9 vs. 52.3), which we attribute to fundamental differences in mechanism: Prompt-based methods (like MetaEoL) are designed to elicit latent capabilities from general-purpose, non-fine-tuned LLMs, acting as a surface-level guidance at inference time. On the contrary, our PDF framework is a holistic, training-time idea. It combines (A) deep prefix injection to create parallel pathways with (B) a MIM objective to explicitly enforce diversity. This synergy pervasively adapts the model's computation during training, compelling it to develop a more robust and generalizable representation space. It is this deep, framework integration—rather than superficial, inference multi-prompting—that is responsible for the superior embedding quality.
>   3. *From an Efficiency Standpoint*, our method achieves gains with no additional inference cost (1x), while prompt-based methods (like MetaEoL) incur a significant 8x computational overhead.
>
> - **Conclusion:** The above comparisons validate that our complete PDF framework—the combination of deep injection and MIM-based regularization during training— is the true source of our performance gains, and represents a more effective and efficient solution than surface-level prompting for training high-quality embedding models.

---

> ### Comment · Reviewer_A3ti · 2025-11-26
>
> Thank you for the rebuttal. I still have concerns.
>
> 1. The MetaEoL comparison doesn’t seem very informative. MetaEoL is meant for prompt-steerable, general-purpose LLMs, not for a contrastively fine-tuned embedding model like VLM2Vec, which isn’t really responsive to prompts. So its failure in this setting is expected, and doesn’t show that multi-prompting itself is ineffective or unrelated to the gains you see with PDF.
>
> 2. The “diversity” between the two paths still looks very limited. The paths produce almost the same representations, and aggregation gives no improvement. This makes the diversity look more like a mild regularization effect rather than meaningful complementary behavior, which raises questions about how essential the parallel decoupling design actually is.
>
> These issues make it difficult to fully support the claimed mechanism.

---

> > ### Author Response · Authors · 2025-11-26
> >
> > We thank you for the continued discussion and your thoughtful follow-up comments. We hope that our detailed response below, including new experimental results, will fully address your remaining concerns.
> >
> > **On the MetaEoL Comparison and Multi-Prompting Effects**
> >
> > We thank the reviewer for this insightful point. We agree that there is a model-method mismatch in applying MetaEoL, a framework designed for general-purpose LLMs, to our fine-tuned embedding model. Therefore, its performance degradation is expected.
> >
> > Our initial goal with that experiment was to test the hypothesis of a 'generic multi-prompting effect.' While the setup had limitations, as you rightly pointed out, it demonstrated that naively applying diverse, unseen prompts at inference time is not a silver bullet for specialized models. This stands in contrast to our framework, which integrates diversity as a core, end-to-end objective during training.
> >
> > To directly address your question about general-purpose models, we conducted a new experiment on the base, non-fine-tuned Qwen2VL-2B model. The result is shown as the table below.
> >
> > | Method             | IND  | OOD  | MEAN |
> > |:-------------------|:----:|:----:|:----:|
> > | Qwen2VL-2B         | 3.1  | 1.8  | 2.6  |
> > | Qwen2VL-2B-MetaEoL | 3.5  | 0.7  | 2.3  |
> > | PDF-VLM2Vec-2B     | 69.5 | 56.9 | 63.9 |
> >
> > This new result provides a crucial insight: even on a general-purpose MLLM, naive multi-prompting fails to provide any meaningful benefit for universal multimodal retrieval task. Note that, MetaEoL was originally designed for pure text retrieval tasks, which involve shorter context lengths compared to multimodal retrieval tasks that include long sequences of image tokens. Hence, one plausible explanation is that Qwen2VL itself performs poorly on this general multimodal retrieval task, so adding MetaEoL yields no improvement.
> >
> > Therefore, our work, which focuses on a novel training-time strategy, addresses this fundamental challenge and holds significant value for advancing MLLM-based embedding models.
> >
> > **On the Necessity of the Parallel Decoupling Design**
> >
> > We thank the reviewer for this insightful framing. We agree that our mechanism can be understood as a form of regularization. However, we would respectfully argue against the characterization of this effect as "mild." It is a powerful, structured regularizer that is fundamentally enabled by, and cannot be separated from, our core architectural design: the parallel paths and the MIM decoupling objective. The significant and consistent performance improvements demonstrated across different model architectures and scales (e.g., 2B and 7B models) serve as direct evidence of its effectiveness and robustness. This confirms that our approach is not a marginal tweak but a potent and generalizable training strategy.
> >
> > The necessity of this design is best demonstrated by two key pieces of evidence:
> > 1. Direct Causal Evidence from Ablation:
> > The most direct proof of its importance comes from the ablation study in our main paper. When we remove the MIM loss (i.e., disable the decoupling mechanism), the model's overall performance drops by a significant 1.2 points. This directly demonstrates that the parallel decoupling design is not a "mild" or incidental effect, but an essential component responsible for a substantial portion of our performance gains.
> > 2. Mechanistic Evidence of Decoupling:
> > While the final embeddings must be similar to solve the primary contrastive task (leading to high cosine similarity), the MIM objective successfully forces the internal computational paths to remain distinct. As shown in Figure 3(b) of our paper, the MIM loss effectively reduces the similarity between prefix tokens. This internal diversity forces the shared model backbone to learn a more robust and generalized data manifold, preventing it from overfitting to a single perspective.
> >
> > In summary, we posit that the "limited" diversity in the final output is a feature. The goal isn't to create complementary experts for aggregation, but to use MIM-enforced diversity as a powerful regularizer during training. The 1.2-point performance drop when this regularizer is removed is the clearest evidence of its necessity.

---

> ### Comment · Reviewer_A3ti · 2025-11-26
>
> Thank you for the further response and the new experiments. However, I still find the MetaEoL comparison difficult to interpret because the baseline Qwen2VL-2B base model does not have instruction-following ability, which makes its performance on MMEB extremely weak. It is therefore unclear whether the conclusions about multi-prompting are meaningful in this setting.
>
> Regarding the diversity mechanism, I appreciate the additional discussion, but I still remain unconvinced. The final representations remain highly similar, and the evidence provided so far does not fully support the necessity of the parallel-path design.
>
> I will keep my original score.

---

> > ### Author Response · Authors · 2025-11-27
> >
> > We sincerely thank you for your continued engagement and for pushing us to strengthen our claims. We understand your remaining skepticism, particularly regarding the fairness of the MetaEoL comparison and the necessity of our diversity mechanism.
> >
> > To address these valid concerns, we conducted a new, comprehensive ablation study designed specifically to isolate the effects of each component of our framework against fair, "MetaEoL-like" baselines trained under identical conditions.
> > We hope these new results will finally resolve your concerns.
> >
> > **New Ablation Study: PDF vs. MetaEoL-like Variants**
> >
> > We designed three "MetaEoL-like" variants on top of VLM2Vec-Qwen2VL-LR and compared them against our full PDF model. All models were trained with the same learning rate and number of iterations.
> > 1. *Variance 1*: 4 distinct learnable prompts (20 tokens) for input tokens . Train: Randomly select 1 prompt per batch. Infer: Average all 4 paths.
> > 2. *Variance 2*: 2 distinct learnable prompts (20 tokens) for input tokensn. Train: Process 2 paths in parallel (like PDF). Infer: Aggregate all 2 paths.
> > 3. *Variance 3*: Same as Variant 2, but with our MIM loss added.
> >
> > | Method                   | IND  | OOD  | MEAN |
> > |:-------------------------|:----:|:----:|:----:|
> > | Variance 1               | 60.2 | 52.6 | 56.8 |
> > | Variance 2               | 60.7 | 52.5 | 57.0 |
> > | Variance 3               | 61.7 | 53.6 | 58.1 |
> > | PDF-VLM2Vec-Qwen2VL-LR   | 62.8 | 53.3 | 58.6 |
> >
> > **Analysis and Key Takeaways:**
> >
> > This new study provides clear, quantitative evidence for our claims by allowing direct, apples-to-apples comparisons:
> > - PDF Outperforms Fair Multi-Prompt Baselines:Our full PDF model (58.6) significantly outperforms both Variant 1 (56.8) and Variant 2 (57.0). This demonstrates that our method's superiority is not an artifact of an unfair comparison; even when MetaEoL-like strategies are fairly implemented and trained on the same backbone, our integrated framework is more effective.
> > - Evidence for the Necessity of MIM Loss:This is the most critical comparison. By adding our MIM loss to Variant 2, we create Variant 3. The result is a substantial performance jump from 57.0 to 58.1 (+1.1 points). This directly and causally proves that the MIM decoupling objective is not a "mild" or incidental effect. It is a crucial component that provides significant gains, even when applied to a standard multi-prompt framework.
> > - Evidence for the Benefit of Deep Prefix Injection: Finally, comparing our full model (Ours, 58.6) with Variant 3 (58.1) isolates the effect of our injection strategy. The +0.5 point gain shows that our deep prefix injection is more effective at steering the model's representations than using standard multi-prompt prefixes, even when both are regularized by the same MIM loss.
> >
> > In conclusion, this new, detailed ablation study systematically dismantles the system to prove the value of each component. It confirms that our performance gains stem from a synergistic combination of a powerful MIM decoupling regularizer and an effective deep injection mechanism.
> >
> > We are truly grateful for your rigorous feedback, which directly inspired this deeper analysis and has substantially improved the quality and completeness of our experimental validation. We sincerely hope these new results have earned your support.

---

### Official Review · Reviewer_pMBJ · 2025-10-30

**Soundness:** 3
**Presentation:** 3
**Contribution:** 3
**Rating:** 6
**Confidence:** 3

**Summary:**

This paper introduces the Parallel Decoupling Framework (PDF): a single MLLM backbone conditioned by learnable deep prefixes to produce multiple parallel embeddings for one input. Diversity is enforced via Mutual Information Minimization (vCLUB estimator) while each path also receives a contrastive objective. At inference, a single forward pass yields an aggregated embedding. PDF improves MMEB results on VLM2Vec-LLaVA-1.6 (+8.9 points, 7B) and Qwen2VL (+4.2 at 2B; +3.1 at 7B), with efficiency gains.

**Strengths:**

- Parallel prefix-conditioned paths + explicit MI minimization within MLLMs.
- Gains across tasks/scales, including compute-reduced settings.
- Clear framing (SSC→SPP) and conceptual diagrams.
- Practical recipe for better MLLM embeddings with negligible inference overhead.

**Weaknesses:**

- Possible instability/bias of vCLUB MI estimation; lack of analysis on estimator variance and its effect on training.
- Limited ablation on #paths, prefix depth/placement, and aggregation design.
- Generalization beyond MMEB (e.g., long-form retrieval/RAG latency-quality tradeoffs) not explored.

**Questions:**

- How does performance scale with number of parallel paths and MI weight?
- Does MI minimization ever hurt semantic alignment (failure cases)?
- Can the aggregated embedding be replaced by path-specific retrieval (multi-vector index), and what are latency implications?

---

> ### Author Response · Authors · 2025-11-23
> **A). Stability and Variance of vCLUB MI Estimation**
>
> We appreciate the reviewer for pointing out the potential risk of instability in MI estimation. To empirically prove the robustness of our method, we conducted four independent training runs with PDF-VLM2Vec-Qwen2VL-LR, each using a different random seed.
> 1. **Low Variance in Training**: As illustrated in the figure 4 (a) in the appendix, the MIM loss curves across multiple runs exhibit high consistency. The shaded region (representing the variance) is extremely narrow, suggesting that the vCLUB estimator achieves reliable convergence throughout the training process, free from significant fluctuations or instability.
> 2. **Stability in Final Performance**: Our method achieved a final performance of 58.6 ± 0.09 (overall score) across these four runs. This negligible standard deviation confirms that our variational estimation of Mutual Information is stable and does not introduce harmful bias or variance that adversely affects the model's convergence or final embedding quality.

---

> ### Author Response · Authors · 2025-11-23
> **B) More Ablations on Paths, Prefixes, MI weight, and Aggregation**
>
> We appreciate the reviewer's suggestion for more extensive ablations. In the Appendix Table 6, we have carried out stuides on path number (N=2,4), prefix length (10,20,40), and MIM loss weight ($λ_{MIM}$). Below, we add additional results to fully address concerns regarding path scaling, prefix depth, and aggregation design. All experiments are conducted with PDF-VLM2Vec-Qwen2VL-LR.
>
> **1. Scaling Number of Paths (N)**
>
> | Path Number | IND  | OOD  | ALL  |
> |:-----------:|:----:|:----:|:----:|
> |      2      | 62.8 | 53.3 | 58.6 |
> |      4      | 63.0 | 53.0 | 58.6 |
> |      8      | 62.4 | 53.0 | 58.2 |
>
> We extended our study to include N=8. As shown in the table above, performance does not scale linearly with the number of paths. While N=4 offers slight gains on IND tasks, N=8 degrades performance. A deeper analysis reveals the underlying reasons for this result:
> - **Combinatorial Explosion in the MIM Training Objective**: The number of pairwise comparisons required by our MIM loss grows quadratically with the number of paths, N, following the function N(N−1)/2. For N=2/4, this results in just 1/6 pairs, while for N=8, there are 28 pairs that must be optimized. This dramatically increases the complexity of the two-stage adversarial-like optimization, making it substantially more difficult to find a stable equilibrium that balances contrastive leanring with inter-path diversity.
> - **Consequent Representation Collapse**: As N increases, the optimization becomes challenging, leading the model to a "lazy" solution where paths become similar. This phenomenon is empirically validated by our analysis of the average cosine similarity between paths, as presented in the figure above, that is, higher similarity for N=8. The model fails to learn diverse representations, making additional paths redundant.
> - **Suboptimal Fusion of Redundant Paths**: As N increases, path aggregation will be overwhelmed by a large number of similar representations. This may result in attention weights being diluted by redundant information, making it difficult to focus on the useful few paths, ultimately reducing the quality of final results.
>
> **2. Aggregation Design**
>
> We considered two classic vanilla baselines: Mean Pooling aggregation and Path-Specific retrieval (in which each path is supervised independently without fusion, implementing multi-vector indexing). The table below compared our learnable MLP-Softmax aggregation against these two baselines. Results show that MLP-Softmax outperforms Mean Pooling by a margin of 0.6 points.
>
> Latency Analysis: 1. **Aggregation (Single-Vector Search)**: Our method produces a single embedding vector. Retrieval involves a standard, highly-optimized Approximate Nearest Neighbor (ANN) search in a single-vector index. This is the industry standard for low-latency retrieval. 2. **Path-Specific (Multi-Vector Search)**: This approach requires searching with N query vectors against an index where every item is also represented by N vectors. In a practical system, this leads to: 1). N-fold increase in index size and memory. 2). N-fold increase in retrieval computation, as the system must effectively perform N searches or a much more complex aggregated distance calculation. This results in significantly higher query latency.
>
> | Method                   | IND  | OOD  | ALL  |
> |:-------------------------|:----:|:----:|:----:|
> | Path-Specific retrieval  | 61.8 | 52.6 | 57.7 |
> | Mean Pooling aggregation | 61.9 | 53.1 | 58.0 |
> | MLP-Softmax aggregation  | 62.8 | 53.3 | 58.6 |
>
> **3. Prefix Depth and Placement**
> - Placement: The causal attention mechanism within the LLM backbone necessitates prepending the prefix to the sequence. This beginning positioning ensures the prefix is visible to every subsequent token, thereby maximizing its influential scope. As a result, it functions as a global instruction, guiding the semantic direction of the entire sequence from the outset.
> - Depth: We investigated the effect of injection depth by comparing two cases: "Shallow" (injecting into the first layer only) and "Deep" (injecting into all layers). As shown in the table below, results confirm that deep injection is crucial for effectively steering the large-scale massive parameters of the MLLM.
>
> | Injection Depth | IND  | OOD  | ALL  |
> |:----------------|:----:|:----:|:----:|
> | Shallow         | 59.5 | 47.3 | 54.0 |
> | Deep            | 62.8 | 53.3 | 58.6 |
>
> **4. Sensitivity of MI Weight $λ_{MIM}$**
>
> In the Appendix Table 6, we empirically demonstrating that $λ_{MIM}$ is somewhat sensitive. $λ_{MIM}=1e−4$ significantly outperforms 1e−3 (weakened the contrastive task, too much noise) and 1e−5 (insufficient decoupling, similar paths), validating the need for a balanced regularization strength.

---

> ### Author Response · Authors · 2025-11-23
> **C). Generalization beyond MMEB**
>
> We appreciate the suggestion to further explore generalization, particularly in long-form and RAG-like scenarios. We demonstrate the superiority of PDF through three types of evidence:
>
> **1. Strong Generalization on MMEB OOD Split:**
>
> We chose MMEB as the primary benchmark specifically because it includes a dedicated Out-Of-Distribution (OOD) split containing 16 datasets not seen during training. As shown in Table 1, our method achieves significant gains on this OOD split (+4.6% for 2B, +4.3% for 7B), indicating strong capability to generalize to novel/unseen domains.
>
> **2. Long-Form Retrieval & RAG Capabilities (Zero-Shot):**
>
> To address your concern about long-form retrieval (a proxy for RAG performance), we point to our zero-shot evaluation on ShareGPT4V and Urban1K benchmarks in Section 4.2 (Line323, Table 2). These benchmarks are particularly relevant as they are characterized by highly detailed, long-form captions that simulate the complex documents typically found in RAG systems.
> Performance: Our PDF-VLM2Vec-2B achieves a massive +13.2% improvement on Urban1K and +5.3% on ShareGPT4V compared to the baseline. The 7B model also shows consistent gains (+2.9%/+2.3%). This confirms that our "diverse exploration" paths significantly aid in capturing the nuanced semantics required for long-context retrieval.
> Latency-Quality Trade-off: Regarding the RAG latency concern, since our inference uses a single path (detailed in Table 4), we achieve performance gains with zero additional latency compared to the standard VLM2Vec, offering an optimal Pareto frontier for RAG systems.
>
> **3. Additional Experiments on Unseen Datasets:**
>
> To further validate generalization beyond MMEB, we conducted new zero-shot experiments on Fashion200K (e-commerce domain) and InfoSeek (information seeking). As shown in the table below, our PDF-VLM2Vec consistently the VLM2Vec baseline across different model scales and query modalities (Image-Text, Text-Image, etc.).
>
> | Model             | Image-Text  | Image+Text-Text | Text-Image  | Image+Text-Image+Text |
> |:------------------|:-----------:|:---------------:|:-----------:|:---------------------:|
> |                   | Fashion200K | InfoSeek        | Fashion200K | InfoSeek              |
> |                   | (R@10)      | (R@5)           | (R@10)      | (R@5)                 |
> | VLM2Vec-Qwen2VL-2B        | 4.7         | 30.4            | 5.2         | 41.5                  |
> | PDF-VLM2Vec-Qwen2VL-2B    | 7.7         | 33.9            | 12.7        | 41.4                  |
> |---|---|---|---|---|
> | VLM2Vec-Qwen2VL-7B        | 9.6         | 37.2            | 13.6        | 44.7                  |
> | PDF-VLM2Vec-Qwen2VL-7B    | 10.2        | 42.1            | 14.1        | 44.6                  |

---

> ### Author Response · Authors · 2025-11-23
> **D). Does MI minimization ever hurt semantic alignment?**
>
> Insightful question. If the penalty on mutual information is excessive, the model might generate task-irrelevant noise just to satisfy the MIM constraint, thereby posing a risk of damaging semantic alignment. However, in our PDF framework, we mitigate this risk through two mechanisms, and we have explored the failure boundary empirically:
> - Theoretical Safeguard (the "Anchor"): As detailed in Section 3.3, we apply the Contrastive Loss not just to the aggregated embedding, but also to each individual parallel path. This acts as a strong semantic anchor, compelling every path to independently learn a valid representation that aligns with the target, ensuring that diversity does not come at the cost of relevance.
> - Empirical Failure Case: We explicitly explored this trade-off in Table 6.
>   - Failure Case: When we increased the MIM weight $λ_{MIM}$ to 1e-3, the performance dropped from 58.6 to 57.3. This confirms your intuition: excessive diversity pressure can compromise alignment.
>   - Optimal Balance: By setting the MIM weight to 1e-4, we strike a balance where the contrastive loss maintains alignment, while MIM serves as an effective regularizer to broaden semantic coverage.

---

> ### Comment · Reviewer_pMBJ · 2025-11-26
>
> Thank you to the authors for the detailed response, which has addressed most of my concerns. I will keep my current score.

---

> > ### Author Response · Authors · 2025-11-27
> >
> > Thank you for your positive feedback and for acknowledging our efforts. We are glad to have addressed most of your concerns and appreciate your continued support.

---

### Author Response · Authors · 2025-12-01
**Summary of Review and Rebuttal Process**

This comment provides a summary of the review and rebuttal process for our submission, #1211. The paper initially received scores of 6,4,6, with reviewers recognized our work for introducing a novel and effective Parallel Decoupling Framework (PDF) while raising several constructive points. Our rebuttal systematically addressed the following concerns through extensive new experiments and analysis.

1. **Demonstrated Broader Generalization and Competitiveness.** In response to concerns about generalization, we expanded our evaluation to include more diverse model backbones and datasets. Crucially, we demonstrated that our method remains highly competitive even against a SOTA-like model trained on a massive amount of additional synthetic data, confirming the robustness and practical value of our approach.
2. **Strengthened Core Design Choices via Ablations.** We conducted a battery of new ablation studies to validate our design. These included experiments on a higher number of parallel paths, different aggregation methods, and an analysis of vCLUB's stability, solidifying the rationale behind our framework's architecture and confirming its training stability.
3. **Systematically Validated the Necessity and Superiority of Our Core Mechanisms.** The validity of our MetaEoL comparison and the necessity of our parallel decoupling design were the primary points of contention with Reviewer A3ti. We addressed this through a decisive, final ablation study specifically designed to settle these two key questions:
   - To Settle the "Validity" Question: Instead of using the original MetaEoL on trained VLM2Vec models, we designed fair, "MetaEoL-like" variants trained from scratch on the same fine-tuned backbone as PDF-VLM2Vec. This created a true "apples-to-apples" comparison, removing the "instruction-following" confounder and establishing a valid baseline to test against.
   - To Settle the "Necessity" Question: We then used these new baselines to precisely quantify the contribution of our core components:
      - Quantifying the Necessity of MIM Loss: We proved the decoupling loss is not a "mild effect" by showing that adding it to an otherwise identical parallel-prompt model yielded a significant +1.1 point gain. This causally demonstrates its necessity.
      - Quantifying the Superiority of Deep Injection: We further showed that our deep prefix injection is superior to shallow prompting by demonstrating a +0.5 point gain over the strongest MetaEoL-like variant.

This rigorous process successfully convinced Reviewer pMBJ and Reviewer 9b5b, who acknowledged that their concerns were addressed and maintained their positive scores. While Reviewer A3ti maintained their score before the discussion period closed, our final and most decisive ablation study was conducted in direct response to their last comment. We believe this final experiment provides the definitive evidence needed to resolve their remaining skepticism.

In summary, the rebuttal process has substantially improved the paper, making its core claims more robustly validated than ever. We are confident in its contribution and thank you for your time and consideration.

---

### Meta-Review · Area_Chair_mnJo · 2026-01-13

**Summary:**

This paper proposes the Parallel Decoupling Framework (PDF), which introduces parallel prefix-conditioned paths within a single MLLM backbone and enforces diversity through mutual information minimization. The approach is motivated, clearly presented, and shows promising gains on the MMEB benchmark across multiple backbones, with minimal inference overhead.

However, after carefully weighing the reviews, I recommend reject, which I agree with reviewer A3ti.

While two reviewers are positive, both are moderate-confidence (confidence 3) and note limitations in generalization and evaluation scope. In contrast, the critical review (confidence 4) raises core technical concerns that remain insufficiently addressed. Most notably, the paper does not convincingly demonstrate that the proposed parallel paths actually yield meaningfully diverse semantic embeddings. Key evidence is missing: there is no analysis of attention behavior, feature subspace separation, or per-path performance. Moreover, the near-identical results between single-prefix and aggregated inference strategies raise reasonable doubt about whether diversity is effectively realized or exploited in practice.

In addition, the novelty relative to existing prompt- or prefix-based embedding methods is not clearly isolated. The absence of direct comparisons to strong and relevant baselines (e.g., prompt-based multi-embedding methods such as MetaEoL) makes it difficult to attribute the observed gains specifically to deep prefix injection and mutual information minimization, rather than to more generic multi-prompt effects. The evaluation is also heavily concentrated on MMEB, limiting confidence in the framework’s generality across other retrieval or embedding benchmarks.

Overall, while the idea is promising and the results encouraging, the paper currently lacks the mechanistic evidence, ablations, and comparative analysis needed to substantiate its central claims. These issues go beyond minor presentation gaps and instead concern the core validation of the method’s effectiveness and novelty.

**Reviewer Concerns:**

Two are solved, and one still has the concerns.

**Reviewer Scores:**

No scores have been changed.

---

### Decision · Program_Chairs · 2026-01-26

Reject